# SatBird: Bird Species Distribution Modeling with Remote Sensing and Citizen Science Data

**Mélisande Teng**
Mila, Université de Montréal
tengmeli@mila.quebec

**Amna Elmustafa**
Mila, Stanford University
amna.elmustafa@mila.quebec

**Benjamin Akera**
Mila, McGill University
akeraben@mila.quebec

**Yoshua Bengio**
Mila, Université de Montréal
yoshua.bengio@mila.quebec

**Hager Radi Abdelwahed**
Mila
hager.radi@mila.quebec

**Hugo Larochelle**
Mila, Google Research, Google Deepmind
hugolarochelle@google.com

**David Rolnick**
Mila, McGill University
drolnick@cs.mcgill.ca

## Abstract

Biodiversity is declining at an unprecedented rate, impacting ecosystem services necessary to ensure food, water, and human health and well-being. Understanding the distribution of species and their habitats is crucial for conservation policy planning. However, traditional methods in ecology for species distribution models (SDMs) generally focus either on narrow sets of species or narrow geographical areas and there remain significant knowledge gaps about the distribution of species. A major reason for this is the limited availability of data traditionally used, due to the prohibitive amount of effort and expertise required for traditional field monitoring. The wide availability of remote sensing data and the growing adoption of citizen science tools to collect species observations data at low cost offer an opportunity for improving biodiversity monitoring and enabling the modelling of complex ecosystems. We introduce a novel task for mapping bird species to their habitats by predicting species encounter rates from satellite images, and present SatBird[1], a satellite dataset of locations in the USA with labels derived from presence-absence observation data from the citizen science database eBird, considering summer (breeding) and winter seasons. We also provide a dataset in Kenya representing low-data regimes. We additionally provide environmental data and species range maps for each location. We benchmark a set of baselines on our dataset, including SOTA models for remote sensing tasks. SatBird opens up possibilities for scalably modelling properties of ecosystems worldwide.

## 1 Introduction

Climate change is a major driver of biodiversity loss, affecting ecosystem services which, in turn, impacts various human wellbeing aspects [32, 45]. It is crucial to understand the changing distributions of species globally to inform policy decisions, for example in shaping land use and land conservation choices. However, there remain large knowledge gaps about species distribution, due principally to the amount of effort and expertise required for traditional field monitoring. Furthermore, mapping species habitats at scale can be challenging as habitats have varying scales depending on species.

---

[1]Project Website: https://satbird.github.io/

37th Conference on Neural Information Processing Systems (NeurIPS 2023) Track on Datasets and Benchmarks.

For example, song sparrows are found in a wide variety of open habitats, including tidal marshes, agricultural fields, overgrown pastures, forest edges, and suburbs whereas Kirtland's warblers only breed in young jack pine forests. Traditional methods for species distribution models (SDMs) use environmental data to predict the distribution of species across geographic space. However, they generally focus either on narrow sets of species or narrow geographical areas, while species interact with each other, and multi-species modelling is needed for understanding ecosystems. The limitations of traditional methods lie in their computational cost, the insufficient ability of models to account for complex relationships between different variables, and the limited availability of the type of data used [50, 47], at a fine-grained resolution.

Machine learning algorithms for remote sensing have increasingly seen wide applicability across sustainability-related domains (see e.g. [55]) and have been suggested as a promising tool for SDMs [8]. Moreover, the surge of data collection on citizen science platforms along with improved data quality validation processes in recent years offers tremendous opportunity for scientific research, as species observation records from these sources can cover a larger temporal and geographic extent at a finer resolution and at a lower cost than traditional sampling methods. Indeed, citizen science data and remotely sensed ecosystem attributes such as vegetation indices have been shown to improve the performance of models, especially for less widespread species [4]. Recently, the GeoLifeCLEF challenge [41] was introduced with the goal of directly predicting plant and animal abundance from 1m-resolution aerial images, using deep computer vision. However, the GeoLifeCLEF benchmark has proven extremely challenging, potentially since only one species (out of 17,000 overall) is associated with each location, even though numerous other species could likely be found in the same place.

We propose to use remote sensing to infer the joint distribution of many species for a given location, using publicly available citizen science observation records as ground truth. Our approach leverages the fact that a species' presence or absence at a location depends on the ecosystem present there, and therefore the abundances of different species are highly correlated. We present SatBird, a dataset and benchmark for the task of jointly predicting the encounter rates of bird species. Encounter rates are a widely used measure in species distribution modeling, and correspond to the probability of an observer to encounter this species if they visit a given location.

Our contributions include:

- Proposing and framing the task of predicting bird species encounter rates jointly at a specific location using remote sensing data (Section 2)
- We introduce SatBird, a dataset for the above task, obtained from publicly available bird observation and satellite data sources (Section 4), composed of the following sub-datasets: (i) USA summer dataset, generally corresponding to the breeding season, (ii) USA winter dataset, the nonbreeding season, (iii) Kenya dataset, as an example of a low-data regime.
- We benchmark a variety of popular models on SatBird and show the efficacy of deep computer vision methods for this task (Section 5).

We release the dataset and code for dataset preparation and benchmark making it possible to easily extend the dataset to other regions in the world.

## 2   Task definition

We propose to predict bird encounter rates from remote sensing data with the goal of completing species distribution mapping in places that have not been surveyed. We use bird sighting records from the citizen science database eBird [36], which has 80 million records of almost all 10000 global bird species. In particular, we leverage observation reports called *complete checklists*. Each location (termed a *hotspot*) in the eBird database is associated to a number of complete checklists containing all species a birdwatcher was able to observe at a specific date and time. Complete checklists are *presence-absence data* –reflecting both the presence of reported species and the absence of non-reported species – and therefore may be considered almost as informative as expert field surveys. If $h$ is a hotspot, and $s_1, \ldots, s_n$ the species of interest, then our goal is to build a machine learning model that takes as input a satellite image of $h$ (and optionally other data) and predicts the vector $\mathbf{y}^h = (y_{s_1}^h, ..., y_{s_n}^h)$, where $y_s^h$ is the number of complete checklists reporting species $s$ at $h$ divided by the total number of complete checklists at $h$.

This ratio $y_s^h$ can be understood as an *encounter rate*, the probability for a visitor to observe a species if they visit the same location (hotspot). We aim to jointly predict this quantity for all relevant bird

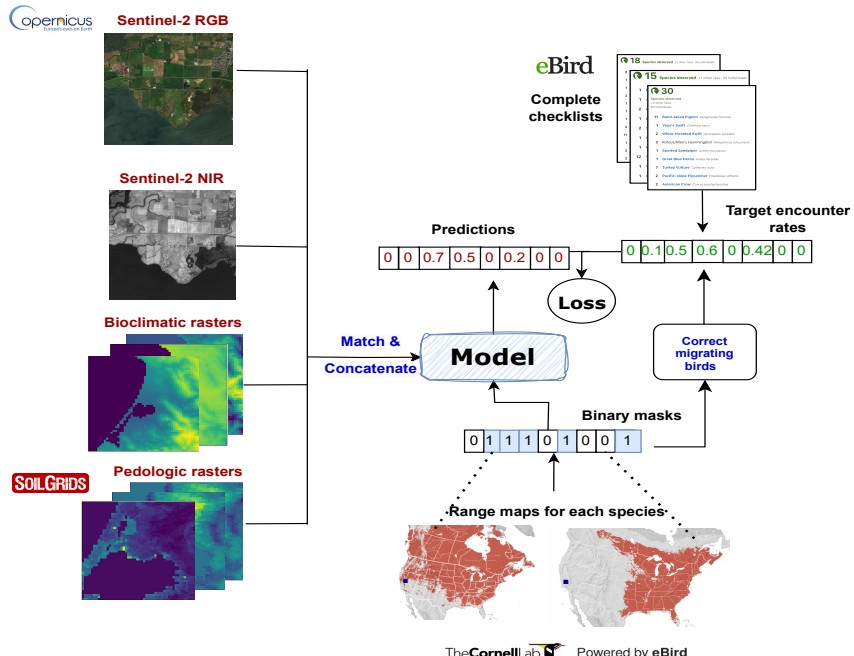

Figure 1: **An overview of data streams for SatBird**, as well as inputs and outputs for the task for predicting species encounter rates. Sentinel-2 10m-resolution satellite data can be used along low resolution environmental data as input to a model after matching their resolutions. Labels are derived from eBird complete checklists. Observations of vagrants (migrating birds) in the labels are corrected with range maps from eBird, which can also be incorporated in the model to make it geography-aware.

species. Thus, our task can be considered as a supervised multi-output regression problem. We chose to predict encounter rates as they are an important quantity in ecology, and they are also used extensively in the form of "hotspot bar charts" in the eBird platform in order to summarize the species in a given location for birdwatchers and ornithologists. These bar charts are used to suggest to users what species may be expected in a particular location. However, they rely only on past data, and the suggestions cannot be extended to places that have not yet been visited. Moreover, encounter rates from eBird have not been modeled jointly to take the interaction between species in the same habitat into consideration, which is important to inform species presence estimates. Our goal is to bridge this gap by modeling the encounter rates of many species jointly and predict them in locations with few or no observations. An overview of the interaction between our data and models is shown in Figure 1.

## 3 Related Work

There have been an increasing number of citizen science initiatives to collect species observation data for a variety of taxa in the past decade, from butterflies, to plants, to sharks [26, 58, 51]. Indeed, by crowdsourcing data collection efforts, it is possible to gather data not only over a larger temporal and geographic extent but also at a fine resolution [16]. As a result, the number of papers using citizen science for SDMs has increased at approximately double the rate of the overall number of SDM papers [22]. **eBird** data has been increasingly used for scientific research on birds, from assessing climate change-driven vulnerability of species, to modelling population change to predicting virus transmission [59, 35, 12]. The eBird status and trends project [24] from the eBird team combines satellite images with raw eBird data and uses statistical models and machine learning to build visualizations and tools to better understand migration, abundance patterns, range boundaries, and other patterns around the world. Other large scale citizen science databases used for SDM include **iNaturalist** [31] in which users record observations of species by taking geolocated pictures. In total, it has 113 million species observations. However, because it is limited to observations with pictures, it is more prone to sampling biases, both geographically and towards certain species (e.g. larger,

more brightly coloured species [9, 10]). It is also *presence-only* – that is, while there is data on the presence of certain species at a given location, it is assumed that other species may also be present.

Remote sensing data has been used for a variety of biodiversity monitoring applications, including predicting land cover classes for downstream ecological modeling [3], measuring the size of groups of animals [63], identifying tree species from crowns [61], and localizing bird nesting sites [29]. Bioacoustics has been used for bird monitoring [2] but one of the main limitations relates to the detectability of species, for example in densely populated areas with many anthropogenic sound sources [25]. Therefore, we will focus on discussing related work with remotely sensed imagery. In the **GeoLifeCLEF 2020 challenge [14]**, they proposed a task combining remote sensing and citizen science data, where the goal is to predict the localization of plant and animal species using 1.6M geo-localized observations from France and the USA of 17,000 plant and animal species from aerial images and environmental features. The labels are derived from citizen science databases of presence-only species observations and each location is associated with only one species, limiting the ability to accurately model multiple species at a time. A newer version (2023 edition) of this classification challenge proposes to focus only on plant species and to use Sentinel-2 satellite images rather than aerial images. While each location is still associated with one species, a validation set with presence-absence labels is provided, highlighting the importance of presence-absence data, at least for the evaluation of such species modeling tasks. Methods deriving information from satellite images have also been developed in the context of avian studies but they usually use a suite of carefully selected measures calculated from the imagery rather than leveraging its full potential [5, 21].

A particular field of application that saw the development of large remote sensing images datasets is agriculture, with an emphasis on the temporal resolution for crop monitoring [28, 44, 62]. Recently, a number of methods have been introduced to learn robust representations of remote sensing data, with the goal of using them for a variety of downstream tasks [44, 52, 53]. In particular, MOSAIKS [55] proposes to use a single encoding of satellite imagery for diverse prediction tasks, and Satlas [7] and SatMAE [15] provide pre-trained models on a large number of remote sensing images, arguing that they can be useful feature extractors for remote sensing tasks.

Compared to existing datasets and methods for SDM, our proposed approach has several advantages:

- We fully leverage presence-absence data from a large citizen science database and compute encounter rates such that biases due to the different number of observers visiting locations are mitigated (see Section 2).

- We enable modelling of a wide range of species at a time across large geographical areas. To our knowledge, this is the first attempt at predicting encounter rates for many species jointly, building on recent advances in machine learning for remote sensing.

- Our proposed pipeline makes it easily extendable to other regions in the world as all data sources are open; eBird [36] data is available in all countries in the world and satellite Sentinel-2 data is publicly available.

## 4   SatBird Dataset

We introduce SatBird, a dataset for the task of predicting encounter rates defined in Section 2 from remote sensing images and environmental data with labels derived from eBird observation reports, with the continental USA and Kenya as regions of interest. Observations in eBird are particularly skewed towards North America, and we choose the USA as a first region of interest, given the extensive and reliable data available on bird observations. We consider the summer season (June-July) and the winter season (December-January) separately. For most North American bird species, the summer reflects breeding season range, while the winter reflects nonbreeding range. Note that we omit the spring and fall migration seasons, in which species distributions may vary dramatically, and which are also often of less interest to ecologists than breeding habitat.In Kenya, we consider the distribution of birds all-year-round as migration is negligible.

In this section, we detail the data sources and data preparation for SatBird for the USA summer and winter seasons and Kenya. Table 1 summarizes the data for each of the considered subsets *SatBird-USA-summer*, *SatBird-USA-winter* and *SatBird-Kenya*. Figure 1 summarizes the different sources of data for SatBird.

|  | USA-summer | USA-winter | Kenya |
|---|---|---|---|
| Number of hotspots | 122593 | 53361 | 9975 |
| Number of species | 670 | 670 | 1,054 |
| Satellite RGBNIR reflectance | ✓ | ✓ | ✓ |
| Satellite true color image | ✓ | ✓ | ✓ |
| Bioclimatic rasters | ✓ | ✓ | ✓ |
| Pedologic rasters | ✓ | ✓ | |
| Range maps | ✓ (586 species) | ✓ (620 species) | |

Table 1: Summary of the data provided for each of the SatBird subsets: USA-summer, USA-winter, and Kenya.

## 4.1 Bird species data

We used complete checklists from eBird [36]. For USA, we extracted checklists recorded from 2010 to 2023 at hotspots in the continental USA in June-July and December-January for the summer and winter seasons respectively. We filtered out hotspots with fewer than 5 complete checklists recorded, to ensure more meaningful encounter rates. We also excluded locations in the sea using the 5-meter resolution US nation cartographic boundaries provided by the Census Bureau's MAF/TIGER [48]. We included all regularly occurring species in the region, as denoted by ABA Codes 1 and 2 [1], which include regular breeding species and visitors. Code-1 species are more widespread, while Code-2 species can have restricted range or occur in lower density, bur are still widespread. We omit species found only in Hawaii and Alaska, as well as one Code-2 seabird species with rare oceanic observations, leaving a total of 670 species.

For Kenya, we extracted all complete checklists recorded from 2010 to 2023, limiting ourselves to the 1,054 species found regularly in Kenya according to Avibase [6]. Due to the scarcity of data in this region, we applied no criteria regarding the minimum number of checklists per hotspot or the month in which an observation was made. There are also a larger number of non-migratory birds present in Kenya than the USA, further motivating our choice to consider records aggregated across seasons.

An additional preparation step was merging hotspots with different IDs in eBird that had the same latitude and longitude, pooling the checklists together, unless all checklists were reported by the same observer on the same day, in which case we dropped the hotspot. We then computed the encounter rates for each hotspot as explained in Section 2 and corrected for vagrants (species seen in hotspots outside of their geographical range) by using range maps from eBird to set the target encounter rates to zero in these hotspots when they were available. We aggregated the species observations over 13 years, only considering seasonal change in distributions in the USA and leaving annual temporal change for future iterations of this work. As we aggregated data per hotspot, we did not take into account the reporting patterns of birders. Previous work [34] demonstrates that estimating the observer expertise can improve species distributions from eBird data. This may also be considered for future iterations of this work.

## 4.2 Environmental data

Following the GeoLifeCLEF 2020 dataset [14], we extracted 19 bioclimatic variables rasters of size $50 \times 50$ and $6 \times 6$ for the USA and Kenya datasets respectively and of spatial resolution approximately 1 km centered on each hotspot from WorldClim 1.4. Bioclimatic variables are often used to model species distributions as a function of climate [33]. They represent annual trends for temperature, precipitation, solar radiation, wind speed and water vapor pressure. Additionally, for the USA dataset, we also extracted 8 pedologic (soil) variables with resolution 250 meter from SoilGrids [30], which provides global soil properties maps, including pH, soil organic carbon content and stocks. These maps are obtained with machine learning methods trained on soil profile observations and environmental covariates derived from remote sensing. We provide more details on the environmental variables in Appendix A.

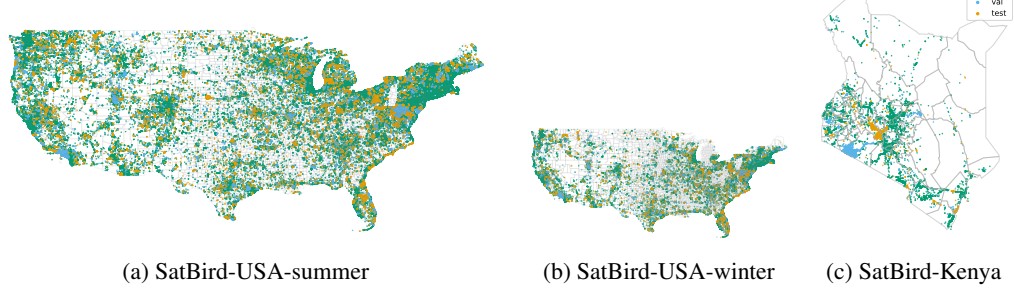

|  |  |  |
|:---:|:---:|:---:|
| (a) SatBird-USA-summer | (b) SatBird-USA-winter | (c) SatBird-Kenya |

Figure 2: Distribution of hotspots across the training, validation, and test sets.

### 4.3 Satellite data

For each hotspot, we extracted data at 10 meter resolution for RGB and NIR reflectance data from Sentinel-2 satellite tiles covering a square of about 5 km$^2$ centered around the hotspots. Additionally we also extracted the true color image RGB bands. We extracted images with cloud coverage of at most 10%, keeping the least cloudy image in the time windows (June 1 - July 31, 2022) for USA-summer, (December 1, 2022 - January 31, 2023) for USA-winter, and (January 1, 2022 - January, 1 2023) for Kenya datasets respectively. In the case where no image was found for a hotspot with these criteria, we composed mosaics with images with up to 25% cloud coverage in the considered time window - and taking the median values. We associate one image per hotspot, considering that a recent satellite image is best representative of our species data, since there are more checklists in recent years compared to earlier years.

### 4.4 eBird range maps

We extracted range maps from the eBird Status and Trends [24] for the summer season in the USA from eBird by filtering ranges between June and July 2021 which are available for 586 of the 670 species considered in the USA. For the winter dataset, we were able to extract range maps for 620 out of 670 species using the same process but filtering ranges between December and January using the R package ebirdst [17]. For each hotspot $l$, we construct a binary mask $m_l$ of size 670 where $m_l^{(i)} = 1$ if $l$ is within the range of species $i$. Range maps were not available for Kenya data.

### 4.5 Aligning data

Finally, we aligned the satellite and the species observation data to obtain the final set of hotspots included in the SatBird dataset, as some satellite images were not found (for example if the cloud cover criterion was not met). Only hotspots with a corresponding satellite image with height and width greater than 128 pixels were kept, with their associated the environmental data rasters. In this way we obtained 122,593 hotspots for the USA summer season, 53,361 for the USA winter season and 9,975 in Kenya. In summary, each hotspot is associated with the following input data: 1 satellite image with RGB & NIR reflectance (4) + RGB true color bands (3), bioclimatic rasters (19), and pedologic rasters (8). The targets are computed from the aggregated checklists, and correspond to species encounter rates. Figure 3 shows the distribution of the number of species per hotspot.

### 4.6 Dataset splits

In order to account for spatial auto-correlation and overfitting that can arise from random splits of geospatial data [54], we use the `scikit-learn` DBSCAN algorithm [19] implementation to cluster hotspots. DBSCAN finds core samples and expands clusters from them, and we specify core samples to have at least 2 hotspots with a maximum allowed distance between them of 5km. We obtained 12,650 clusters which were then randomly assigned to train, validation and test splits such that they respectively contained 70%, 15%, and 15% of the hotspots for the US summer dataset. We kept the same split assigment for winter US locations also present in the summer US locations, assigned the

rest in a 70 %, 15 %, 15% repartition in the train, validation and test sets. The same procedure was followed for the data in Kenya. Figure 2 shows the spatial distribution of hotspots for the dataset and we provide number of hotspots per split in Appendix A.

## 5 Benchmarking

In this section, we present the baselines used to benchmark our task, predicting species encounter rates. We choose a wide range of baselines covering simple regression trees, deep CNNs and transformers.

### 5.1 Models

**Mean encounter rates:** We propose a first basic baseline which is simply the average of encounter rates over the training set for each species.

**Environmental baseline:** Following the GeoLifeCLEF [41] challenge, we propose an environmental baseline, using Gradient Boosted Regression Trees on the bioclimatic and pedological variables extracted at each of the hotspots.

**MOSAIKS [55]:** This model was proposed as an accessible method that generalizes well across a wide range of tasks at significantly lower computational cost compared to training a deep neural network for each task. It performs a one-time unsupervised image featurization by convolving a set of randomly selected patches with the input satellite image. These features can then be reused for an unlimited number of tasks. We propose to use the MOSAIKS method to extract features from our satellite images dataset and combine them with the environmental variables before training a regressor to predict the encounter rates at each hotspot.

**SATLAS pretrained transformer [7]:** The literature suggests pretraining Swin-v2 [40] transformer on a large number of remote sensing images to target distribution shifts in remote sensing tasks. The model is pre-trained on Satlas, a dataset of 1.3 million satellite images from different sources, and is proven to perform well in both in-distribution and out-of-distribution settings. They provide weights for models pre-trained on low-resolution and high resolution images. We freeze the Satlas parameters and train a final fully connected layer to predict our target.

**SatMAE [15]:** This is an unsupervised pre-training framework based on Masked Autoencoders (MAE) for tasks related to remote sensing, specifically for temporal or multi-spectral satellite imagery data. It has been shown to perform well on supervised and transfer learning tasks. We apply transfer learning to our dataset using SatMAE pre-trained model on the fMoW [13] dataset. We use the pre-trained model as a feature extractor and train a dense layer to predict the species encounter rates.

**ResNet-18 [27]:** CNNs have been widely used as feature extractors for remote sensing imagery. We use ResNet-18 [27] architecture, using the RGB and NIR bands reflectance data as input and we initialize the network with ImageNet pretrained weights. Since ImageNet has only RGB bands, the initialization of the first layer for the NIR band is performed by sampling from a normal distribution parameterized by the mean and standard deviation of the layer's weights for the other bands. We also experiment with RGB true color images as an input and compare the results against reflectance data.

### 5.2 Including geographical information

We propose to make the CNN baseline model spatially explicit to account for the geographical ranges of some species. For example, Eastern and Western wood pewees have the same habitat but are found in different regions and do not co-occur. A model blind to geographical information might predict both species in the same place, which is undesirable when producing maps for conservation planning, and could also impede training. We propose to use **range maps** from eBird, constructing binary masks to zero-out the predicted encounter rates of species when a given location falls outside of their geographical range. For location $l$, the range mask $r_l$ is a vector of size equal to the number of bird species, and $r_l^i = 0$ if $l$ is outside of species $i$ geographical range, and $r_l^i = 1$ otherwise. If the range map is not available for a species, the predictions are left untouched ($r_l^i = 1$ for all species). For species $i$, if $z_l^i$ is the logit, the final prediction is then $\hat{y}_l^{(i)} = \sigma(z_l^i) * r_l^{(i)}$.

## 5.3 Evaluation metrics

Beyond optimization for common regression metrics (MSE, MAE), it is desirable for the predicted most likely species in a given hotspot to coincide with those that have been observed most frequently. In fact, while we model many species jointly, our targets are zero-inflated due to the absence of species at many hotspots, which is a frequent challenge in SDM [46, 38] . We therefore report top-10 and top-30 accuracies, representing for $k = 10$ and $k = 30$ the number of species present in both the top $k$ predicted species and the top $k$ observed species, divided by $k$. However, the number of species observed varies considerably across hotspots (Figure 3), unlike in the GeoLifeCLEF challenge for which there is only one ground truth species for each hotspot. We therefore also define an *adaptive top-k accuracy* as the top-$k$ value where $k$ is the total number of species observed at that hotspot.

## 6  Experiments

### 6.1  Experimental details

In our experiments, we consider a region of interest of $640$ m$^2$, center-cropping the satellite patches to size $64 \times 64$ around the hotspot and normalizing the bands with our training set statistics. When adding environmental data, we match their resolution to that of the satellite images. Random vertical and horizontal flipping, random noise, blur, contrast and brightness were used for data augmentation. Models are trained with cross entropy loss: $\mathcal{L}_{CE} = \frac{1}{N_h} \sum_h \mathcal{L}_h = \frac{1}{N_h} \sum_h \sum_{s(\text{species})} -y_h^s \log(\hat{y}_h^s) - (1 - y_h^s) \log(1 - \hat{y}_h^s)$, where $N_h$ is the number of hotspots $h$, $y$ the predictions and $\hat{y}$ the ground truth encounter rates.

Satlas and SatMAE models use the true color images as input. They do not support the use of environmental data. The input is further processed by normalizing its values between 0 and 1. For MOSAIKS, we extracted 1024 features for each true color image, combined them with the environmental data and trained an XGBoost regressor on them. For the ResNet-18-based models, we experiment with different inputs: RGB true color images, RGBNIR reflectance values, and RGBNIR reflectance values combined with the bioclimatic and pedologic data. We normalize the bioclimatic and pedologic data variable-wise with training set statistics, aligning them to the satellite images' resolution and stacking the corresponding patches to the images.

All models were trained with batch size 128 using Adam [37] optimizer. Each experiment is run on 3 seeds and results report the average of 3 seeds on the test set. Training, validation and test sets follow the splitting described in Section 4. All ResNet-18 models were trained for 50 epochs. Satlas and SatMAE were trained for 100 epochs, freezing the pretrained model and training only the last layer. More details about experiments including resources used are available in Appendix B.

### 6.2  Results & discussion

In this section, we present results for all our baseline models using SatBird-USA-summer and SatBird-USA-winter shown in Tables 2 and 3 respectively. Validation set results for both datasets are reported in Appendix C.We also report baselines for SatBird-Kenya in Appendix D.

We observe that models trained with satellite images only, either the true color image or the reflectance values, do not outperform the environmental baseline. However, combining satellite images with environmental rasters results in a significant improvement, in all metrics, with the highest improvement in the top-30 (13%), top-k (4%), and top-10 (8%) metrics in SatBird-USA-summer and improvement in the top-30 (10%), top-k (4%), and top-10 (5%) metric for SatBird-USA-winter, compared to the environmental baseline. This highlights the value of satellite data for our task, and also the importance of using both environmental and remote sensing data. The models using environmental data along with either RGBNIR reflectance images or RGB visual images achieve nearly-equal competitive results, but our best models use RGB images, along with environmental data, titled ResNet-18 (img+env) in Tables 2 and 3. We suggest incorporating other information as input such as land cover and altitude data which can be obtained from publicly available sources [18, 11] for further improvements.

For the transformer-based models, both Satlas and SatMAE do not outperform ResNet-18 when applied to the true color image only. This observation suggests that we may need further finetuning for these models on our data rather than freezing the model weights. We note that Satlas outperforms

| Model | MAE[1e-2] | MSE[1e-2] | Top-10 | Top-30 | Top-k |
|---|---|---|---|---|---|
| Mean encounter rates | $3.1 \pm 0.0$ | $0.9 \pm 0.0$ | $26.9 \pm 0.0$ | $38.6 \pm 0.0$ | $44.8 \pm 0.0$ |
| Environmental baseline | $2.3 \pm 0.0$ | $0.6 \pm 0.0$ | $42.9 \pm 0.1$ | $57.8 \pm 0.1$ | $63.8 \pm 0.1$ |
| Satlas (img) | $2.9 \pm 0.01$ | $0.9 \pm 0.0$ | $30.5 \pm 0.1$ | $46.1 \pm 0.0$ | $48.6 \pm 0.1$ |
| SatMAE (img) | $3.0 \pm 0.0$ | $0.9 \pm 0.0$ | $28.9 \pm 0.1$ | $43.5 \pm 0.1$ | $46.5 \pm 0.2$ |
| MOSAIKS (img+env) | $2.5 \pm 0.0$ | $0.7 \pm 0.0$ | $41.9 \pm 0.0$ | $56.7 \pm 0.0$ | $62.2 \pm 0.0$ |
| ResNet-18 (img) | $2.6 \pm 0.02$ | $0.8 \pm 0.0$ | $35.5 \pm 0.1$ | $52.2 \pm 0.2$ | $54.5 \pm 0.2$ |
| ResNet-18 (refl) | $2.6 \pm 0.01$ | $0.8 \pm 0.0$ | $36.1 \pm 0.1$ | $53.3 \pm 0.1$ | $55.5 \pm 0.1$ |
| ResNet-18 (img+env) | $2.2 \pm 0.02$ | $\mathbf{0.64} \pm 0.0$ | $\mathbf{46.3} \pm 0.2$ | $\mathbf{65.5} \pm 0.2$ | $\mathbf{66.9} \pm 0.1$ |
| ResNet-18 (refl+env) | $\mathbf{2.1} \pm 0.01$ | $0.65 \pm 0.0$ | $45.8 \pm 0.1$ | $65.1 \pm 0.2$ | $66.6 \pm 0.1$ |
| ResNet-18 (refl+env+RM) | $2.16 \pm 0.0$ | $0.65 \pm 0.0$ | $45.6 \pm 0.2$ | $65.1 \pm 0.1$ | $66.7 \pm 0.1$ |

Table 2: **Test results on the SatBird-USA-Summer**: Best results are shown in bold. *img* refers to using the RGB true color image, *refl* refers to using RGBNIR reflectance bands, *env* refers to using the environmental data (bioclimatic and pedologic variables). *RM* refers to the use of range maps.

| Model | MAE[1e-2] | MSE[1e-2] | Top-10 | Top-30 | Top-k |
|---|---|---|---|---|---|
| Mean encounter rates | $2.5 \pm 0.0$ | $0.7 \pm 0.0$ | $27.6 \pm 0.0$ | $45.7 \pm 0.0$ | $51.4 \pm 0.0$ |
| Environmental baseline | $1.8 \pm 0.0$ | $0.4 \pm 0.0$ | $48.7 \pm 0.0$ | $63.7 \pm 0.0$ | $68.5 \pm 0.0$ |
| Satlas (img) | $2.3 \pm 0.0$ | $0.7 \pm 0.0$ | $31.6 \pm 0.1$ | $51.5 \pm 0.1$ | $54.1 \pm 0.1$ |
| SatMAE (img) | $2.4 \pm 0.0$ | $0.7 \pm 0.0$ | $28.6 \pm 0.8$ | $50.2 \pm 0.1$ | $52.6 \pm 0.1$ |
| MOSAIKS (img+env) | $1.9 \pm 0.0$ | $0.5 \pm 0.0$ | $47.8 \pm 0.0$ | $62.1 \pm 0.0$ | $66.4 \pm 0.0$ |
| ResNet-18 (img) | $2.2 \pm 0.2$ | $0.6 \pm 0.0$ | $35.3 \pm 0.4$ | $54.3 \pm 0.2$ | $56.9 \pm 0.4$ |
| ResNet-18 (refl) | $2.2 \pm 0.0$ | $0.6 \pm 0.0$ | $37.1 \pm 0.9$ | $56.5 \pm 0.9$ | $58.7 \pm 0.9$ |
| ResNet-18 (img+env) | $1.69 \pm 0.0$ | $\mathbf{0.4} \pm 0.0$ | $\mathbf{51.1} \pm 0.3$ | $\mathbf{69.5} \pm 0.24$ | $70.9 \pm 0.2$ |
| ResNet-18 (refl+env) | $\mathbf{1.67} \pm 0.0$ | $0.4 \pm 0.0$ | $50.9 \pm 0.1$ | $69.3 \pm 0.0.1$ | $70.8 \pm 0.1$ |
| ResNet-18 (refl+env+RM) | $1.69 \pm 0.0$ | $0.4 \pm 0.0$ | $51.0 \pm 0.1$ | $69.4 \pm 0.1$ | $\mathbf{71.0} \pm 0.1$ |

Table 3: **Test results on the SatBird-USA-winter**: Best results are shown in bold. *img* refers to using the RGB true color image, *refl* refers to using RGBNIR reflectance bands, *env* refers to using the environmental data (bioclimatic and pedologic variables). *RM* refers to the use of range maps.

SatMAE, likely because the former was pre-trained on a very large dataset of Sentinel-2 images which is the same source (and has the same resolution) as our images. For SatMAE, the model was pretrained on the fMoW dataset with a higher resolution (0.5m) than the images in our dataset. MOSAIKS also performs reasonably well compared to other deep models (ResNet-18 for example) for the true color image and environmental raster input, considering that it is a computationally lightweight model.

We also observe in Tables 2 and 3 that using RGBNIR reflectance bands as input yields better results than using true color images (while not using environmental data), given the information added by the NIR channel. This aligns with previous conclusions [4] that remote sensing indices using the NIR band, such as the NDVI improve the performance of SDMs. Moreover, adding range maps, does not hurt the model but it does not seem to add a significant contribution. Range maps can still be useful in cases where the data is not as abundant as SatBird-USA-summer. In particular, we still recommend to use this approach whenever available with smaller-sized datasets. We had experiments with using latitude and longitude as inputs to a location encoder, concatenating their features with the model features but we found lower performance for this baseline (see Appendix B.5).

# 7 Conclusion

In this work, we present and release SatBird, a large-scale dataset for species distribution modeling using remote sensing and citizen science data. SatBird includes datasets in the USA for summer and winter seasons and one dataset in Kenya. We detail the process for building SatBird, as well as using it with different models. We also present benchmark results for the dataset on a wide range of popular remote sensing algorithms. Two limitations of the present work include using a single satellite image to represent a season, while the landscape can change over time, and that satellite images are extracted from one recent year, while eBird data is aggregated across multiple recent years.

We are planning a next version of SatBird that will include multiple satellite images for each hotspot to account for changes over time. We also aim to expand SatBird using citizen science data for other organisms, not merely birds (this is more challenging since most such datasets are presence-only).

As SatBird is intended to directly impact biodiversity monitoring, we look forward to integrating the approaches we have introduced into existing tools for ecology and policy. One immediate application for models trained on SatBird is eBird's existing tool that lists the "likely species" in a given area, to be available for poorly monitored locations. This tool currently relies on encounter rates from past checklists and is therefore only available in well-monitored locations. Models trained on SatBird could estimate such encounter rates for poorly monitored locations via remote sensing. We hope such input will be valuable to researchers seeking to understand biodiversity and climate change, as well as policymakers interested in evaluating conservation priorities across different areas of land.

## Acknowledgments and Disclosure of Funding

This research was enabled and funded by Mila (mila.quebec). This project was supported by the Canada CIFAR AI Chairs program and a Google-Mila grant. The authors grateful to Sal Elkafrawy for their contribution in adapting the MOSAIKS baseline code and to the Mila IDT team for technical support.

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

# A Further Dataset Description

## A.1 Number of species per hotspot

A well-known issue in species distribution modeling when looking at many species at a time is the zero-inflated nature of the targets, which also appears in our dataset. Indeed, all 670 considered species in the USA and 1054 species in Kenya are never found in the same place together. Figure 3 shows the distribution of the number of species with non-zero encounter rates per hotspot in the SatBird-US-summer and the SatBird-Kenya datasets.

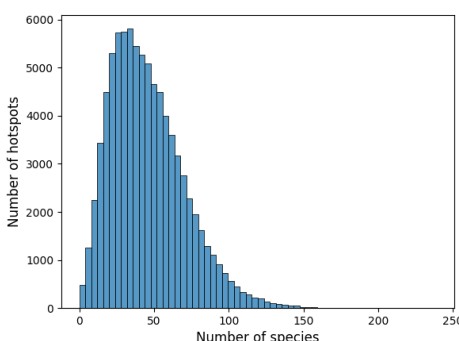 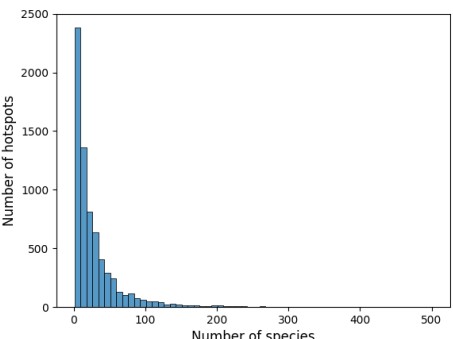

Figure 3: Distribution of the number of species encountered per hotspot for the USA-summer (left) and Kenya (right) training sets. On average a hotspot in these datasets has resp. 48 and 30 different species reported.

## A.2 Dataset splits

Table 4 describes the number of hotspots present in each split for each dataset of SatBird. The splits were obtained by following the process described in Section 4.

|            | USA summer | USA winter | Kenya |
| ---------- | ---------- | ---------- | ----- |
| train      | 85553      | 38241      | 6988  |
| validation | 18511      | 8323       | 1658  |
| test       | 18529      | 6797       | 1329  |

Table 4: Number of samples in each split for the three datasets

## A.3 Environmental variables

In Table 5, we provide a description of the environmental variables provided in the SatBird dataset. The bioclimatic data is taken from the highest resolution data available (30 seconds resolution, corresponding to about 1km2) from WorldClim [20, 23]. Data for the USA dataset was obtained following the method proposed by [14], and we used WorldClim1.4 which has data aggregated from 1960-1990. For Kenya, we extracted data from WorldClim2.1 which has data aggregated for 1970-2000. The soil data comes from SoilGrids and has resolution 250m [30].

## A.4 Distribution of the targets

Figure 4 shows the distribution of species by average encounter rate over the USA-summer training set hotspots. Most species have a low average encounter rate reflects both the zero-inflated nature of the targets. In Figures 5 and 6, we show the distribution of encounter rates for a very common species all across the USA, the American robin (*Turdus migratorius*) and the California quail (*Callipepla californica*) whose range only extends to the west of the USA. If we compare the two distributions,

| Bioclimatic variables (30 arcsec resolution) | |
|---|---|
| Name | Description |
| bio_1 | Annual Mean Temperature |
| bio_2 | Mean Diurnal Range (Mean of monthly (max temp - min temp)) |
| bio_3 | Isothermality (bio_2/bio_7) (* 100) |
| bio_4 | Temperature Seasonality (standard deviation *100) |
| bio_5 | Max Temperature of Warmest Month |
| bio_6 | Min Temperature of Coldest Month |
| bio_7 | Temperature Annual Range (bio_5-bio_6) |
| bio_8 | Mean Temperature of Wettest Quarter |
| bio_9 | Mean Temperature of Driest Quarter |
| bio_10 | Mean Temperature of Warmest Quarter |
| bio_11 | Mean Temperature of Coldest Quarter |
| bio_12 | Annual Precipitation |
| bio_13 | Precipitation of Wettest Month |
| bio_14 | Precipitation of Driest Month |
| bio_15 | Precipitation Seasonality (Coefficient of Variation) |
| bio_16 | Precipitation of Wettest Quarter |
| bio_17 | Precipitation of Driest Quarter |
| bio_18 | Precipitation of Warmest Quarter |
| bio_19 | Precipitation of Coldest Quarter |
| Pedologic variables (250m resolution) | |
| Name | Description |
| orcdrc | Soil organic carbon content (g/kg at 15cm depth) |
| phihox | Ph x 10 in H20 (at 15cm depth) |
| cecsol | Cation exchange capacity of soil (cmolc/kg 15cm depth) |
| bdticm | Absolute depth to bedrock in cm |
| clyppt | Clay (0-2 micro meter) mass fraction at 15cm |
| sltppt | Silt mass fraction at 15cm depth |
| sndppt | Sand mass fraction at 15cm depth |
| bldfie | Bulk density in kg/m3 at 15cm depth |

Table 5: Description of the bioclimatic and pedologic variables extracted from WorldClim1.4 and SoilGrids.

we see that for the California quail, more hotspots have zero encounter rate which is consistent with the fact that its range is smaller. Table 6 shows in which states the species is found, namely California and neighboring states.

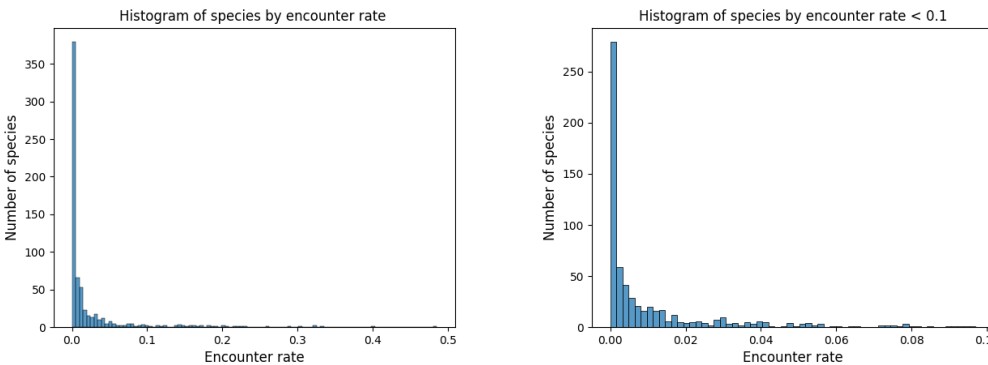

Figure 4: Distribution of species by average encounter rate over the USA-summer training set (left). The figure on the right zooms in the histogram for species with encounter rate < 0.1

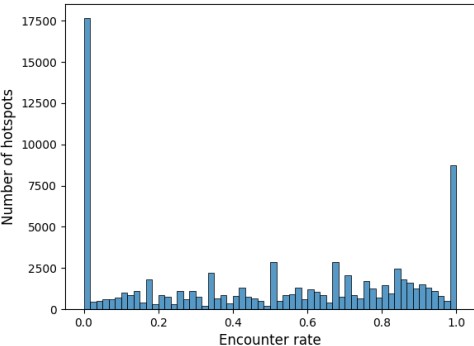

Figure 5: Encounter rates distribution of the American robin across the USA-summer training set

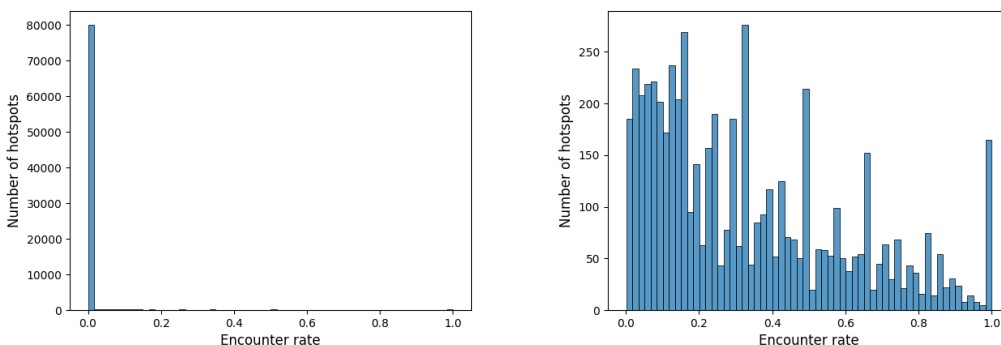

Figure 6: Encounter rates distribution of the California quail across the USA-summer training set. We whos the full distribution (left) and the distribution for hotspots with non-zero encounter rates(right).

## A.5 Dataset format

For each SatBird dataset subset, we provide a `.tar.gz` file which is organized as follows:

```
Dataset name (USA-summer, USA-winter, Kenya)
    |- images/
    |- images_visual/
    |- corrected_targets/
    |- environmental_data/
    |- train, valid, test csv files
    |- species_list.txt (file with the scientific name of the considered species)
```

Each hotspot has one corresponding file in each subfolder. `images` contains the RGBNIR reflectance data in the format of `.tif` file, `images_visual` has the RGB true color images in the format of `.tif` file, `environmemntal_data` has the environmental rasters in the format of `.npy` file, and `corrected_targets` have the encounter rates for each hotspot in the format of `.json` file. We provide more details about the files in the Datasheet for SatBird.

## A.6 Where to find the dataset

We have released a project website, where you can find links to access dataset and code. Our dataset is available for download at this link: `https://drive.google.com/drive/folders/1eaL2T7U9Imq_CTDSSillETSDJ1vxi5Wq?usp=sharing`. The companion code for the dataset

| State | Number of hotspots |
|---|---|
| California | 3248 |
| Washington | 962 |
| Oregon | 921 |
| Idaho | 384 |
| Nevada | 168 |
| Utah | 65 |
| Montana | 10 |

Table 6: Distribution of hotspots in the USA-summer training set per state, for which the encounter rate (target) for the California quail is non-zero.

preparation pipeline and benchmark can be found here: `https://github.com/RolnickLab/SatBird`.

# B Experiments

We have presented simple baselines to lay the foundations for the task of regressing encounter rates, trying to cover different data inputs and different categories of models (Gradient-boosted regression, CNNs, transformers).

We used an internal cluster in our organization that has enough GPUs. For the deep vision ResNet baselines, on Nvidia A100 GPU, one epoch takes 54 minutes maximum (for the model Resnet-18(refl+env+RM)), with batch size of 128 and using 16 GB of RAM memory. Other models of Resnet-18 took less time. There are experiments where we used more powerful GPUs, resulting in less than 1 day for training 50 epochs. Each baseline was trained 3 times on 3 different seeds. As for the Satlas and SatMAE baseline, each experiment took two days to train on one GPU. We used Pytorch framework [49] in the implementation of our models.

For the MOSAIKS baseline, we extracted 1,024 features for each 10m-resolution image from Sentinel-2 and tried both XGBoost and Ridge regression on the combination of the environmental and image features obtained with the Random Kitchen sinks methods proposed in MOSAIKS. The MOSAIKS model [55] was originally trained on around 1x1km 256x256 pixels RGB images (resolution of about 4m/pixel) and 8,192 features were extracted for each of the training 100,000 images. In our case, we found XGBoost to perform better on the SatBird-USA-Summer dataset and Ridge regression to perform better on the SatBird-USA-Winter and SatBird-Kenya datasets. Each model was trained with 3 different seeds and results of each baseline are reported on the mean of the 3 experiments.

Below, we add additional experiments on SatBird-USA-summer including: 1) hyperparameter search, 2) using pretrained weights from another remote sensing dataset, 3) using different loss functions, 4) training seperate models per bird species.

## B.1 Hyperparameter search: learning rate

We conducted a hyperparameter search a posteriori for the learning rate on the ResNet-18 (refl + env + RM) baseline for the SatBird-USA-summer dataset for values in the range of $\langle 1 \times 10^{-4}, 5 \times 10^{-3} \rangle$, and found that a learning rate of $1 \times 10^{-4}$ gave slightly better results. Hence, we used $1 \times 10^{-4}$ in our reported experiments. We show the loss curves for a few models in our hyperparameter search in Figure 7.

## B.2 Initializing with pre-trained weights on a remote sensing dataset

We compare initializing the ResNet-18 (refl + env + RM) model with ImageNet weights (as reported in the paper) with an initialization from Seasonal Contrast weights [44]. We find that the models perform similarly as reported in Table 7.

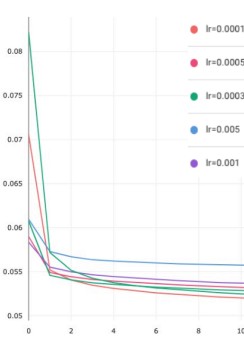
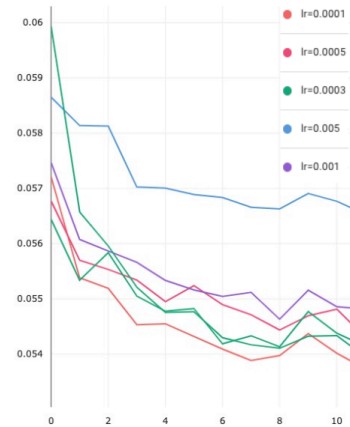

(a) Training loss vs epochs            (b) Validation loss vs epochs

Figure 7: Loss curves for ResNet-18 (refl+env+RM) models trained on SatBird-USA-summer with different learning rates.

| Model | MAE[1e-2] | MSE[1e-2] | Top-10 | Top-30 | Top-k |
|---|---|---|---|---|---|
| ImageNet initialization | 2.16 | 0.65 | 45.7 | 65.1 | 66.8 |
| Seasonal Contrast initialization | 2.2 | 0.65 | 45.6 | 65.1 | 66.6 |

Table 7: **Test results on the SatBird-USA-summer dataset** for ResNet-18 (refl+env+RM) models initialized with ImageNet and Seasonal Contrast weights

## B.3 Comparing different loss functions

Throughout the main paper, our experiments used cross entropy as the loss function as mentioned in Sec 6. We have conducted further experiments using other regression loss functions such as L1 loss and L2 loss. We have also extended our cross entropy loss to focal loss [39] to address the class imbalance. Table 8 shows that cross entropy loss achieves the best accuracies, and even the best MSE over the test set.

## B.4 Could the model be better at predicting only one bird instead of the whole distribution?

Our goal with the Satbird dataset and task is to predict many birds at once, and we expect that having information about other birds allows a single model to learn from features that are common across many bird species and that it may indeed benefit predictions for the less common birds. We hypothesize that training a single model to predict all species is better when modeling species distribution to capture interactions and co-occurrences among species. To verify this, we conduct experiments where we train ResNet-18 model (refl+env+RM) on 20 random species (including the most common and least common species), a model on each of these species (total 20 models) and

| Model | MAE [1e-2] | MSE [1e-2] | Top-10 | Top-30 | Top-k |
|---|---|---|---|---|---|
| Cross Entropy | 2.16 | **0.65** | **45.74** | **65.13** | **66.78** |
| L1 Loss | **1.86** | 0.77 | 43.58 | 60.86 | 61.13 |
| L2 Loss | 2.64 | 0.66 | 44.92 | 64.03 | 65.49 |
| Focal Loss | 2.93 | 0.75 | 39.43 | 57.99 | 60.68 |

Table 8: **Using different loss functions for Resnet-18 baseline**: this table compares the performance of Resnet-18 baseline when using different loss functions. Cross entropy Loss outperforms all other losses.

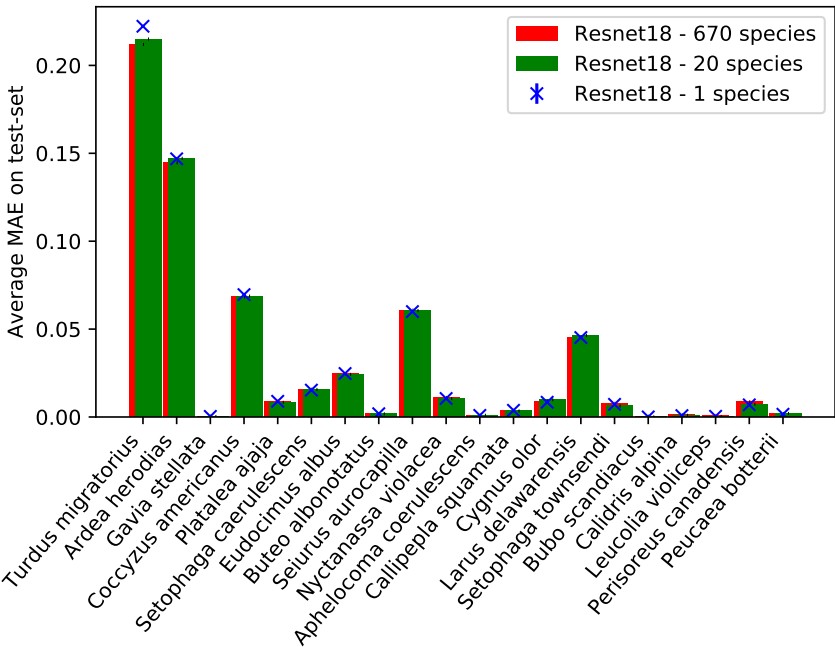

Figure 8: **Training separate models per one species vs. training all species**: *Resnet18 - 670 species* refers to a model trained over all bird species. *Resnet18 - 20 species* refers to a model trained over 20 bird species only. *Resnet18 - 1 species* refers to a model trained over each single species only as specified on the x-axis.

one model on all the 670 species. Then, we compare the predictions of these models over the 20 species to the ground truth and report the mean absolute error (MAE). The difference in performance is, as predicted, quite small; with models trained on more classes (either 20 species or all 670 species) performing slightly better. Figure 8 shows that models trained on 20 or all species can be better than training a model per single species. Even if it may be better to train on less number of species, we suggest that the gain is much less than the compute overhead we will have when training many separate models per species.

## B.5 Incorporating location information

Besides the range map masking method to make the CNN models spatially explicit, we also propose to train a location encoder taking as input latitude and longitude information jointly with the image encoder. We used the same architecture as the one proposed in [42] for the location encoder's architecture, removing the final classification layer, and instead concatenating the obtained features to those obtained from the ResNet image encoder, before passing them to a fully-connected layer to obtain the predicted encounter rates. More specifically, we first map each spatial coordinate $x^l$ (latitude and longitude) to $[\cos(\pi x^l), \sin(\pi x^l)]$. We end up with a input vector of size 4, which is passed through an initial fully connected layer, followed by 4 residual blocks, each consisting of two fully connected layers with a dropout layer in between. The performance of initial experiments with this setting on a subset of the SatBird-USA-summer dataset was not better than what we obtained with range maps masking, and thus we did not explore further this method for this benchmark. However, it could be that the location encoder's architecture or the fusion with the satellite image features could be improved. We show results obtained on a small subset of an early version of the SatBird-USA-summer dataset ( 9k hotspots) which led us to prefer exploring the range maps masking method in Table 9.

| Model | MAE [1e-2] | MSE [1e-2] | Top-10 | Top-30 | Top-$k$ |
|---|---|---|---|---|---|
| Mean encounter rate | $2.9 \pm 0.0$ | $0.7 \pm 0.0$ | $26.45 \pm 0.0$ | $43.91 \pm 0.0$ | $51.46 \pm 0.0$ |
| Env baseline | $2.05 \pm 0.00$ | $0.5 \pm 0.0$ | $43.1 \pm 0.1$ | $62.3 \pm 0.03$ | $68.86 \pm 0.0$ |
| ResNet-18 (refl) | $1.98 \pm 0.00$ | $0.5 \pm 0.0$ | $41.8 \pm 0.3$ | $60.1 \pm 0.2$ | $67.0 \pm 0.0$ |
| ResNet-18 (refl+env) | $1.8 \pm 0.0$ | $0.5 \pm 0.0$ | $48.0 \pm 0.0$ | $67.0 \pm 0.00$ | $72.9 \pm 0.2$ |
| ResNet-18 (refl+env+LE) | $1.7 \pm 0.0$ | $0.5 \pm 0.1$ | $44.5 \pm 0.7$ | $65.0 \pm 0.0$ | $72.0 \pm 0.1$ |
| **ResNet-18 (refl+env+RM)** | $1.8 \pm 0.0$ | $\mathbf{0.5 \pm 0.0}$ | $\mathbf{48.2 \pm 0.2}$ | $\mathbf{67.7 \pm 0.1}$ | $\mathbf{73.4 \pm 0.1}$ |

Table 9: Results for baseline models on the test set of a subset of SatBird-US-summer. *env* indicates the use of environmental variables. *RM* and *LE* refer to the range maps and the location encoder respectively.

## C  Further Analysis on Satbird-USA-summer results

### C.1  Validation set results

We report the performance of baselines on the validation set of the SatBird-USA-summer dataset in Table 10, and SatBird-USA-Winter in Table 11. The results on the validation set for both datasets are consistent with the reported test results.

| Model | MAE[1e-2] | MSE[1e-2] | Top-10 | Top-30 | Top-k |
|---|---|---|---|---|---|
| Mean encounter rates | $3.1 \pm 0.0$ | $0.9 \pm 0.0$ | $27.1 \pm 0.0$ | $38.6 \pm 0.0$ | $44.86 \pm 0.0$ |
| Environmental baseline | $2.3 \pm 0.0$ | $0.6 \pm 0.0$ | $43.1 \pm 0.0$ | $57.7 \pm 0.0$ | $63.9 \pm 0.0$ |
| Satlas (img) | $2.8 \pm 0.0$ | $0.9 \pm 0.0$ | $30.9 \pm 0.1$ | $46.3 \pm 0.0$ | $48.9 \pm 0.0$ |
| SatMAE (img) | $3.0 \pm 0.0$ | $0.9 \pm 0.0$ | $29.4 \pm 0.1$ | $43.8 \pm 0.1$ | $46.9 \pm 0.2$ |
| MOSAIKS (img+env) | $2.5 \pm 0.0$ | $0.7 \pm 0.0$ | $42.1 \pm 0.0$ | $56.6 \pm 0.0$ | $62.2 \pm 0.0$ |
| ResNet-18 (img) | $2.6 \pm 0.0$ | $0.8 \pm 0.00$ | $35.9 \pm 0.29$ | $52.6 \pm 0.1$ | $54.8 \pm 0.1$ |
| ResNet-18 (refl) | $2.6 \pm 0.0$ | $0.79 \pm 0.0$ | $36.2 \pm 0.03$ | $53.8 \pm 0.05$ | $56.01 \pm 0.03$ |
| ResNet-18 (img+env) | $2.1 \pm 0.0$ | $\mathbf{0.6 \pm 0.0}$ | $\mathbf{46.8 \pm 0.14}$ | $\mathbf{66.2 \pm 0.15}$ | $\mathbf{67.6 \pm 0.14}$ |
| ResNet-18 (refl+env) | $\mathbf{2.1 \pm 0.01}$ | $0.64 \pm 0.0$ | $46.4 \pm 0.2$ | $65.9 \pm 0.1$ | $67.3 \pm 0.1$ |
| ResNet-18 (refl+env+RM) | $2.1 \pm 0.01$ | $0.6 \pm 0.0$ | $46.3 \pm 0.12$ | $65.9 \pm 0.1$ | $67.4 \pm 0.0$ |

Table 10: **Validation results on the SatBird-USA-Summer**: Best results are shown in bold. *img* refers to using the RGB true color image, *refl* refers to using RGBNIR reflectance bands, *env* refers to using the environmental data (bioclimatic and pedologic variables). *RM* refers to the use of range maps.

| Model | MAE[1e-2] | MSE[1e-2] | Top-10 | Top-30 | Top-k |
|---|---|---|---|---|---|
| Mean encounter rates | $2.5 \pm 0.0$ | $0.7 \pm 0.0$ | $28.2 \pm 0.0$ | $47.5 \pm 0.0$ | $52.7 \pm 0.0$ |
| Environmental baseline | $1.8 \pm 0.0$ | $0.5 \pm 0.0$ | $49.2 \pm 0.0$ | $63.9 \pm 0.0$ | $68.5 \pm 0.0$ |
| Satlas (img) | $2.3 \pm 0.0$ | $0.7 \pm 0.0$ | $32.5 \pm 0.1$ | $53.2 \pm 0.0$ | $55.4 \pm 0.0$ |
| SatMAE (img) | $2.4 \pm 0.0$ | $0.7 \pm 0.0$ | $29.7 \pm 0.3$ | $51.9 \pm 0.1$ | $54.0 \pm 0.0$ |
| MOSAIKS (img+env) | $2.0 \pm 0.0$ | $0.5 \pm 0.0$ | $47.8 \pm 0.0$ | $62.1 \pm 0.0$ | $66.1 \pm 0.0$ |
| ResNet-18 (img) | $2.2 \pm 0.0$ | $0.6 \pm 0.0$ | $37.6 \pm 0.0$ | $56.3 \pm 0.0$ | $58.6 \pm 0.0$ |
| ResNet-18 (refl) | $2.2 \pm 0.0$ | $0.6 \pm 0.0$ | $38.2 \pm 0.0$ | $57.4 \pm 0.0$ | $59.5 \pm 0.0$ |
| ResNet-18 (img+env) | $\mathbf{1.7 \pm 0.0}$ | $0.4 \pm 0.0$ | $51.45 \pm 0.0$ | $69.59 \pm 0.0$ | $71.0 \pm 0.0$ |
| ResNet-18 (refl+env) | $1.7 \pm 0.0$ | $0.5 \pm 0.0$ | $\mathbf{51.59 \pm 0.0}$ | $\mathbf{69.68 \pm 0.0}$ | $71.2 \pm 0.0$ |
| ResNet-18 (refl+env+RM) | $1.7 \pm 0.0$ | $\mathbf{0.4 \pm 0.0}$ | $51.55 \pm 0.0$ | $69.61 \pm 0.0$ | $\mathbf{71.2 \pm 0.0}$ |

Table 11: **Validation results on the SatBird-USA-Winter**: Best results are shown in bold. *img* refers to using the RGB true color image, *refl* refers to using RGBNIR reflectance bands, *env* refers to using the environmental data (bioclimatic and pedologic variables). *RM* refers to the use of range maps.

| Species scientific name | Training occurrences | MSE ranking | MSE |
|---|---|---|---|
| Calcarius lapponicus | 1 | 1 | 7.46597932e-08 |
| Geopelia striata | 1 | 2 | 7.48139554e-08 |
| Rallus obsoletus | 1 | 3 | 7.48282276e-08 |
| Synthliboramphus scrippsi | 2 | 4 | 7.48671356e-08 |
| Ardenna tenuirostris | 1 | 5 | 7.48679727e-08 |
| Calidris maritima | 4 | 5 | 7.49105737e-08 |
| Spizelloides arborea | 1 | 7 | 7.49142886e-08 |
| Aethia cristatella | 1 | 8 | 7.49201557e-08 |

Table 12: Top predicted species on the USA-summer test set according to MSE. *Occurrences* refers to the number of training hotspots for which the encounter rate (target value) is non-zero.

| Species scientific name | Training occurrences | MSE ranking | MSE |
|---|---|---|---|
| Agelaius phoeniceus | 52630 | 670 | 0.09383779 |
| Turdus migratorius | 68198 | 669 | 0.07773756 |
| Zenaida macroura | 48582 | 668 | 0.07559232 |
| Melospiza melodia | 52107 | 667 | 0.06899277 |
| Hirundo rustica | 48299 | 666 | 0.06486867 |
| Corvus brachyrhynchos | 59977 | 665 | 0.0647007 |
| Geothlypis trichas | 38993 | 664 | 0.0632578 |
| Cyanocitta cristata | 47654 | 663 | 0.05804076 |
| Cardinalis cardinaliss | 48687 | 662 | 0.05713348 |
| Passer domesticus | 41578 | 661 | 0.05654651 |

Table 13: Bottom 10 predicted species on SatBird-USA-summer test set according to MSE in descending order. *Occurrences* refers to the number of training hotspots for which the encounter rate (target value) is non-zero.

## C.2 Analysis of results

On the test set of SatBird-USA-summer, the top 50 species with lowest test-MSE are unsurprisingly species with very low number of occurrences in the training set ($< 11$) (so the model can predict a zero encounter rate and have good MSE for these species). Among those species we find species breeding in arctic regions (e.g. Bombycilla garrulus, Calcarius lapponicus) and seabirds (e.g. Uria lomvia). By design of our dataset, we don't have many observations of seabirds since we excluded all hotspots outside the boundaries of the continental US). Species with largest MSE have $> 30$k hotspots where they were observed in the training set. Tables 12 and 13 show the best and worst predicted species on the USA-summer test set according to MSE.

We show the squared error distribution by target value (encounter rates) for the USA-summer test set in Fig. 10. We binned the target values 19 bins and for each bin averaged the squared errors. The squared error is higher on average for higher target values. In Fig. 9, we show the predicted values against the target values. The target values were binned in 19 bins. The model generally underestimates the target values.

We show the geographic distribution of the top-k error in the predictions for the USA-summer test set in Fig. 11

In Appendix E, we show sample predictions of our benchmark model (ResNet-18 with refl+env+RM) on the SatBird-USA-summer dataset. The example in Fig. 12 shows that the model is able to predict some of the rare species that are very habitat specific commonly found in certain areas, such as the Verdin, which is mainly restricted to scrublands.

## D Results on Kenya dataset

SatBird-Kenya dataset was created with the motivation of transferring a model trained on a large dataset (SatBird-USA-summer) for the task of predicting species encounter rates to a region with less data available. We reported the baselines as done on SatBird-USA.

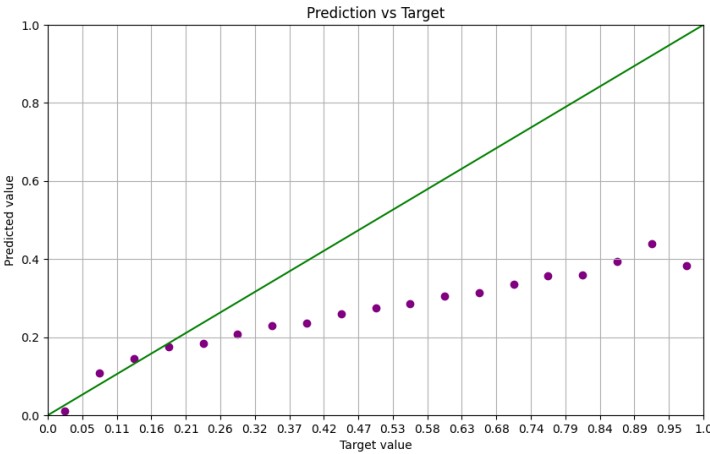

Figure 9: Predicted vs target values (binned in 19 bins) for the USA-summer test set.

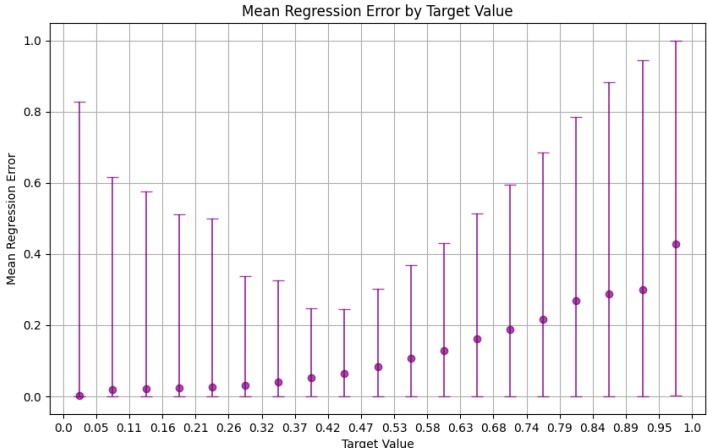

Figure 10: MSE value distribution by target value (binned in 19 bins) for the USA-summer test set. The bars show the minimum and maximum squared error for targets in a given bin.

**Transfer learning:** We experiment with transfer learning from models trained on SatBird-USA data to SatBird-Kenya, a low-data region compared to USA. With a Resnet-18 model trained on SatBird-USA-summer, we transfer the weights and fine-tune on SatBird-Kenya (*finetune-USA*). We also experiment with freezing the weights and only update the last classification layer (*freeze-USA*). All these models are compared to training on SatBird-Kenya from scratch with ResNet-18. We report results for the SatBird-Kenya dataset in Table 14.

**Weighted Loss:** In addition to the above experiments, we tried incorporating the number of complete checklists to the training strategy for Kenya. This was motivated by the observation that Kenya has has relatively fewer hotspots and some hotspots are over represented by the number of complete checklists reported. We address this by balancing the cross entropy loss by the number of complete checklists per hotspot. Our hypothesis is that by balancing the cross entropy by the number of complete checklists per hotspot, our model will learn a more balanced representation of the encounter rates across the whole dataset. eBird, just like many other large biodiversity datasets may have biased estimates of species diversity, as species occurrence generally follow a log-normal

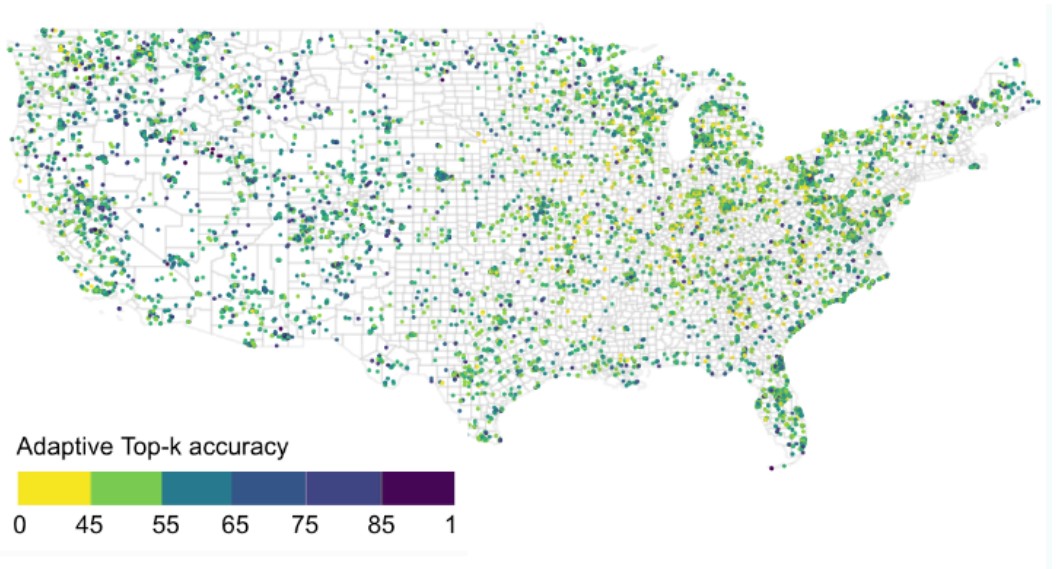

Figure 11: USA-summer test set hotspots colored by adaptive top-k performance for the ResNet-18 (refl+env+RM) model.

distribution with a long tail [56, 57, 43, 60]. Because of this inherent bias, it is unsurprising that the data distribution in Kenya, and parts of the USA will have fewer hotspots with complete checklists.

We define the weighted loss, similar to our original loss, but with an additional weight as follows:

$\mathcal{L}_{WCE} = \frac{1}{N_h} \sum_h w_h \mathcal{L}_h = \frac{1}{N_h} \sum_h w_h \sum_{s(\text{species})} -y_h^s \log(\hat{y}_h^s) - (1 - y_h^s) \log(1 - \hat{y}_h^s)$

where: $N_h$ is the number of hotspots $h$. $w_h$ is the weight corresponding to the number of complete checklists for hotspot $h$. $y$ are the predictions and $\hat{y}$ are the ground truth encounter rates. The summation over $s$ represents a sum over all species.

In Table 14, we note that all ResNet-18 models perform similarly well with the use of cross-entropy as a loss function. One thing to note is that we did not perform any hyperparameter tuning so there is room for improvement for these baselines. Moreover, the sparse nature of the data and the large number of species (1054) to predict, which make this task particularly challenging might also contribute to the limited performance of the models. ResNet-18 initialized with the weights from the model trained on the USA data performed better on some metrics (MSE and MAE), highlighting the relevance of transferring knowledge from one geographical region and set of species to another.
Satlas and SatMAE models seem to perform well overall. The MOSAIKS baseline performed the worst. A first path to improving this baseline would be to experiment with a different number of features generated from MOSAIKS. We also note that the images we considered were 64 * 64 with resolution 10m/pixel, when the original implementation was used for 256*256 images with $\tilde{4}$m/pixel resolution.
As for using weighted loss in ResNet-18 baselines, it did not yield significant improvement to the performance on the Kenya dataset. We think that this is as a result of the fewer number of complete checklists and hotspots overall in Kenya. The task of extending these models to low-data regimes such as in Kenya presents a substantial and important challenge in species distribution modelling. This task is complex, and we think that it would be one direction to expand upon in future work.

# E   Prediction samples

All predictions reported are those obtained with the ResNet-18 (refl+env+RM) model. We sampled randomly hotspots from the SatBird-USA-summer test set and evaluated the model's performance in predicting the top-10 bird species at each location. We selected hotspots from geographically distinct

| Model | MAE[1e-2] | MSE[1e-2] | Top-10 | Top-30 | Top-k |
|---|---|---|---|---|---|
| Mean encounter rates | $3.5 \pm 0.0$ | $1.7 \pm 0.0$ | $14.0 \pm 0.0$ | $19.2 \pm 0.0$ | $23.4 \pm 0.0$ |
| Environmental Baseline | $3.8 \pm 0.0$ | $1.9 \pm 0.00$ | $10.6 \pm 0.1$ | $18.7 \pm 0.0$ | $24.2 \pm 0.1$ |
| Satlas (img) | $3.1 \pm 0.0$ | $1.7 \pm 0.0$ | $12.6 \pm 0.1$ | $29.5 \pm 0.2$ | $23.9 \pm 0.1$ |
| SatMAE (img) | $3.6 \pm 0.2$ | $1.7 \pm 0.0$ | $14.0 \pm 0.8$ | $28.8 \pm 0.4$ | $23.8 \pm 0.1$ |
| MOSAIKS (img+env) | $4.8 \pm 0.0$ | $2.1 \pm 0.0$ | $8.9 \pm 0.0$ | $15.2 \pm 0.0$ | $19.7 \pm 0.0$ |
| ResNet-18(scratch) | $3.9 \pm 0.4$ | $1.7 \pm 0.0$ | $\mathbf{18.5} \pm 1.4$ | $\mathbf{35.1} \pm 0.1$ | $\mathbf{28.6} \pm 0.4$ |
| ResNet-18(finetune-USA) | $3.8 \pm 0.4$ | $1.7 \pm 0.0$ | $17.7 \pm 0.0$ | $34.6 \pm 0.3$ | $28.4 \pm 0.1$ |
| ResNet-18(freeze-USA) | $\mathbf{3.3} \pm 0.1$ | $\mathbf{1.6} \pm 0.0$ | $18.3 \pm 0.2$ | $33.8 \pm 0.4$ | $27.7 \pm 0.1$ |
| WL-ResNet-18(scratch) | $3.6 \pm 0.0$ | $1.7 \pm 0.0$ | $17.4 \pm 0.9$ | $32.7 \pm 1.2$ | $27.2 \pm 0.9$ |
| WL-ResNet-18(finetune-USA) | $3.6 \pm 0.1$ | $1.6 \pm 0.0$ | $17.4 \pm 2.5$ | $33.0 \pm 0.1$ | $27.4 \pm 0.4$ |
| WL-ResNet-18(freeze-USA) | $3.6 \pm 0.1$ | $1.7 \pm 0.0$ | $16.6 \pm 1.5$ | $31.4 \pm 0.3$ | $26.1 \pm 0.3$ |

Table 14: **Test results on the SatBird-Kenya**: *img* refers to using the RGB true color image. All ResNet-18 baselines use *refl+env*, which corresponds to using RGBNIR reflectance bands and the environmental data (bioclimatic and pedologic variables), and WL refers to the weighted Loss. *scratch* refers to training ResNet-18 model from scratch. *finetune-USA* refers to fine-tuning a ResNet-18 model with weights transferred from SatBird-USA-summer. *freeze-USA* refers to using ResNet-18 weights transferred from SatBird-USA-summer while training the last layer only.

regions, featuring different land cover types, including the Pacific Coast (California), Northeast (Maine) and Central US (Ohio). The following figures illustrate the results from these regions.

# F    Authors' statement

The authors bear all responsibility in case of violation of rights.

# G    Licenses

The SatBird Dataset is released under a Creative Commons Attribution-NonCommercial 4.0 International (CC BY-NC 4.0) License (`https://creativecommons.org/licenses/by-nc/4.0/`). Additionally, use of our dataset should comply with the eBird Terms of Use (`https://www.birds.cornell.edu/home/terms-of-use/`) and Terms of Use of the eBird Status and Trends Products (`https://science.ebird.org/en/status-and-trends/products-access-terms-of-use`).

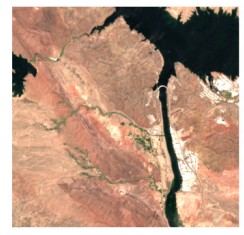

US-CA-L654261

| Species | Predicted probability |
|---|---|
| **Verdin** | 0.410 |
| **Mourning Dove** | 0.394 |
| **Gambel's Quail** | 0.333 |
| **House Finch** | 0.322 |
| Great-tailed Grackle | 0.310 |
| **White-winged Dove** | 0.284 |
| Eurasian Collared-Dove | 0.241 |
| Common Raven | 0.239 |
| Turkey Vulture | 0.239 |
| Black-tailed Gnatcatcher | 0.184 |

| Species | Groundtruth probability |
|---|---|
| Ash-throated Flycatcher | 0.625 |
| Abert's Towhee | 0.5 |
| **Verdin** | 0.5 |
| **Mourning Dove** | 0.375 |
| **White-winged Dove** | 0.375 |
| **House Finch** | 0.375 |
| Ladder-backed Woodpecker | 0.375 |
| **Gambel's Quail** | 0.25 |
| Western Screech-Owl | 0.25 |
| Hooded Oriole | 0.25 |

Figure 12: Hotspot and species at Desilt Wash, San Bernardino Couty in California: Our model correctly identified the Verdin in this region in the top-10 species predicted list. It's worth noting that the Verdin's habitat is a fairly rare species whose habitat is mainly restricted to shrublands.

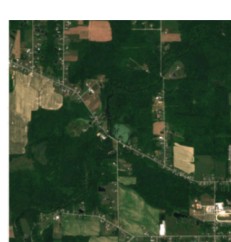

US-CA-L1516413

| Species | Predicted probability |
|---|---|
| **American Robin** | 0.785 |
| **Red-winged Blackbird** | 0.753 |
| **Song sparrow** | 0.727 |
| **Gray Catbird** | 0.686 |
| Northern Cardinal | 0.676 |
| American Goldfinch | 0.603 |
| Common Yellowthroat | 0.556 |
| **Yellow Warbler** | 0.520 |
| **Mourning Dove** | 0.491 |
| American Crow | 0.476 |

| Species | Groundtruth probability |
|---|---|
| **Song Sparrow** | 0.757 |
| **American Robin** | 0.703 |
| **Red-winged Blackbird** | 0.649 |
| Canada Goose | 0.649 |
| Great Blue Heron | 0.622 |
| Eastern Kingbird | 0.540 |
| **Gray Catbird** | 0.513 |
| **Yellow Warbler** | 0.513 |
| Wood Duck | 0.405 |
| **Mourning Dove** | 0.378 |

Figure 13: Hotspot and species in Lampson Reservoir,Ashtabula County, Ohio: Our model predicts the presence of various species in this region, including the song sparrow, red-winged blackbird and gray catbird. These species are consistent with the ground truth and are highly reported on eBird for this location.

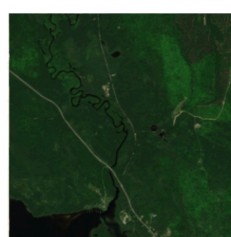

US-ME-L608588

| Species | Predicted probability |
|---|---|
| **White-throated Sparrow** | 0.553 |
| **Northern Parula** | 0.542 |
| Red-eyed Vireo | 0.526 |
| **Black-capped Chickadee** | 0.471 |
| **Red-breasted Nuthatch** | 0.457 |
| Common Yellowthroat | 0.455 |
| **Blue Jay** | 0.449 |
| Swainson's Thrush | 0.431 |
| **American Robin** | 0.414 |
| Magnolia Warbler | 0.414 |

| Species | Groundtruth probability |
|---|---|
| **Northern Parula** | 0.876 |
| **White-throated Sparrow** | 0.826 |
| **Red-breasted Nuthatch** | 0.815 |
| **American Robin** | 0.765 |
| **Blue Jay** | 0.754 |
| **Black-capped Chickadee** | 0.749 |
| Golden-crowned Kinglet | 0.731 |
| Blue-headed Vireo | 0.718 |
| Ovenbird | 0.718 |
| Veery | 0.702 |

Figure 14: Hotspot and species in Kennebago River-Boy Scout Road, Maine: while our model identifies correctly the species with highest ground truth encounter rates, it underestimates the encounter rates values.

