# Datasheet for *SatBird: A Dataset for Bird Distribution Modeling using Remote Sensing and Citizen Science data*

MÉLISANDE TENG, Université de Montréal & Mila
AMNA ELMUSTAFA, Stanford University & Mila
BENJAMIN AKERA, McGill University & Mila
YOSHUA BENGIO, Université de Montréal & Mila
HAGER RADI ABDELWAHED, Mila
HUGO LAROCHELLE, Google DeepMind & Mila
DAVID ROLNICK, McGill University & Mila

You can visit our project website https://satbird.github.io. Our code is publicly available on Github at https://github.com/RolnickLab/SatBird/.

## 1 Dataset Motivation

- **For what purpose was the dataset created?** *Was there a specific task in mind? Was there a specific gap that needed to be filled? Please provide a description.* The dataset was created for the task of predicting bird species encounter rates at scale, from remote sensing imagery and environmental data. Traditional methods in species distribution modelling (SDM) generally focus either on narrow sets of species or narrow geographical areas, while multi-species modelling is needed for understanding ecosystems. SatBird was designed to bridge knowledge gaps in species distributions, by leveraging abundant presence-absence data in the citizen science database eBird and globally available Sentinel-2 satellite data.

- **Who created the dataset (e.g., which team, research group) and on behalf of which entity (e.g., company, institution, organization)?**
  The raw data is obtained from the following sources: eBird [9], Sentinel-2 , SoilGrids [8], and WorldClim [5, 6]. We also obtain data from the eBird status and trends project [7]. The data was processed by the authors of the submission. Additional data from the US Census Bureau (a shapefile of the USA), as well as ABA (American Birding Association) [2] and Avibase data [3] (to select bird species of interest) was used to process the data.

- **Who funded the creation of the dataset?** *If there is an associated grant, please provide the name of the grantor and the grant name and number.* This project was supported by the Canada CIFAR AI Chairs program, a Google-Mila grant and Mila - Quebec-AI Institute.

## 2 Dataset Composition

- **What do the instances that comprise the dataset represent (e.g., documents, photos, people, countries)?** *Are there multiple types of instances (e.g., movies, users, and ratings; people and interactions between them; nodes and edges)? Please provide a description.* The Sentinel-2 data instances are RGB images or reflectance data of the BGRNIR bands centered on the

---

Authors' addresses: Mélisande Teng, Université de Montréal & Mila ; Amna Elmustafa, Stanford University & Mila ; Benjamin Akera,  McGill University & Mila; Yoshua Bengio,  Université de Montréal & Mila; Hager Radi Abdelwahed, Mila; Hugo Larochelle, Google DeepMind & Mila; David Rolnick, McGill University & Mila .

latitude/longitude. The targets are bird species encounter rates obtained by aggregating complete checklists from eBird. The environmental data instances are rasters of bioclimatic and pedological data. Bird range data corresponds to processed range maps from eBird to obtain binary masks for each hotspot. The uploaded dataset only contains the processed instances.

- **How many instances are there in total (of each type, if appropriate)?** The number of data points in the SatBird-USA-summer, SatBird-USA-winter and SatBird-Kenya is 122593, 53361, and 9975 respectively for a total of 185929 data points corresponding to hotspots locations in eBird. For each location, we provide the targets, RGBNIR reflectance data, true color image, environmental rasters data. For the USA datasets we also provide for each location a binary mask of size equal to the number of considered bird species for the season, where the entry is 1 if the location falls within the range of a species, and 0 otherwise. This information was obtained from eBird range maps for the considered species using the breeding and non-breeding season maps for USA-summer and USA-winter respectively. In the case where no range map was available for a given species, all the entries are 1. Additionally we provide files describing the train-validation-test splits proposed in our submission, as well as the list of scientific names of the species considered for the USA and Kenya datasets. Note that SatBird-USA-winter and SatBird-USA-summer data share the same set of species and also have hotspots in common.

- **Does the dataset contain all possible instances or is it a sample (not necessarily random) of instances from a larger set?** *If the dataset is a sample, then what is the larger set? Is the sample representative of the larger set (e.g., geographic coverage)? If so, please describe how this representativeness was validated/verified. If it is not representative of the larger set, please describe why not (e.g., to cover a more diverse range of instances, because instances were withheld or unavailable).*

  The dataset corresponds to a sample of all the hotspots in the eBird dataset in Kenya and in the USA. We explain in the paper the criteria for choosing this sample. While SatBird is representative of the data in eBird, it should be noted that there are inherent spatial biases to the eBird data, and therefore not equal geographical coverage of regions in the USA and Kenya.

- **What data does each instance consist of?** *"Raw" data (e.g., unprocessed text or images) or features? In either case, please provide a description.* We detail the data in each instance below, as well as the processing steps performed to obtain them:
  - **Targets**: .json files, with the entries *"hotspot_id", "probs", "num_checklists"*, where *"probs"* is a list of size equal to the number of considered species containing the target encounter rates and *"num_checklists"* is the number of complete checklists that were aggregated to obtain the encounter rates.
  - **Images**: .tif files corresponding to the BGRNIR reflectance satellite data corresponding to the each hotspot. The reflectance values typically range 0-15000. We recommend using the *tiff* package in Python to read them.

    The images were extracted using Google Earth Engine, specifying cloud coverage of at most 10%, and a region of interest of about 5 km$^2$ centered around the hotspots. We kept the least cloudy image in the time windows (June 1 - July 31, 2022) for USA-summer, (December 1, 2022 - January 31, 2023) for USA-winter, and (January 1, 2022 - January, 1 2023) for Kenya datasets respectively.In some cases, the extracted images were incomplete or missing (in particular at the border between Sentinel tiles). Another round of image extraction was done, this time allowing for composing mosaics with images that have up to 25% cloud coverage in the considered time window - and taking the median values. We then filtered out hotspots for which the obtained satellite tiles had dimensions < 128 * 128 pixels.

- **Visual**: .tif files corresponding to the the true color (RGB) satellite image ( values [0-255]) corresponding to the each hotspot.
- **Environmental**: .npy files, corresponding to the environmental data rasters of size 50x50 for the USA datasets and 6x6 for the Kenya dataset. The choice to extract different sizes of rasters for both countries was motivated by the difference in size of the country and the processing steps to fill NaN values (detailed below). For the USA datasets, the environmental data rasters contain both bioclimatic (19 variables) and pedologic (8 variables) data. For SatBird-Kenya, the environmental raster corresponding to a given location has information for the 19 bioclimatic variables. Bioclimatic data for the USA is extracted from WorldClim1.4 data at 30 seconds (approx. 1 km2) resolution which is the highest available from this database. It has data aggregated for 1960-1990. Soil data is extracted from SoilGrids and has 250m-resolution. We mainly followed GeolifeCLEF[4] 2020 challenge to extract the environmental data; that why we used WorldClim1.4. For Kenya, we upgraded to use WorldClim2.1 for extracting the environmental data.

   Bioclimatic and pedologic data is not available for all locations, and NaN values in the extracted rasters were filled as follows:
   1. Extract rasters with NaN values. Compute the max and min values for each variable over the considered dataset.
   2. Fill NaN values with bilinear interpolation. We used `scipy.interolate.interp2d` function. If interpolation was not possible (less than 2 values in the raster), we left the raster with NaNs,
   3. Clip the values of the rasters with the max and min computed in step 1.
   4. For rasters with remaining NaN values, we filled the values with "point value" means (of non-NaN values) of the the environmental variables obtained when extracting rasters of size 1x1 centered on the hotspots lat-lon. Those "point" values are the ones provided in the train/val/test .csv files and used for the environmental baselines of the benchmark (see description of the .csv files below).

A .pkl file of the **processed range map masks** is provided for USA hotspots for winter and summer, where the columns are the 670 species and the hotspot IDs. In each row, the entry for a species is "True" is the hotspot falls within the species range during the considered season (summer of winter), and "False" otherwise.

We also provide the **list of species** (scientific names) that we consider in the dataset. The ordering in the list is the same one as for the targets.

We also provide **train, test, and validation .csv files** for each dataset. These .csv include the hotspot ID, state and county of the hotspot, bioclimatic (and pedological for the USA) data values. The values for the bioclimatic and pedological data correspond to data extracted in the same way as the environmental rasters, except that a size of 1 (at the lat-lon of the hotspot) was queried. In each .csv, NaN values for the bioclimatic and pedological variables were filled with the mean of the values on the training set. These variables were used for the environmental baselines.

- **What is the format of the data package for each dataset?** Each data package (USA-summer/USA-winter/Kenya) is provided in a zip file as follows (with the content being describe in the previous question)

```
USA_summer / USA_winter / Kenya
|-- train_split.csv
|-- valid_split.csv
```

```
|-- test_split.csv
|-- range_maps.pkl
|-- images/{hotspot_id}.tif
|-- images_visual/{hotspot_id}_visual.tif
|-- environmental_data/{hotspot_id}.npy
|-- targets/{hotspot_id}.json
|-- species_list.txt
```

- **Is there a label or target associated with each instance?** If so, please provide a description. Yes, the targets the encounter rates for the considered bird species at each location. For SatBird-USA, the targets are vectors of size 670 and for SatBird-Kenya they are vectors of size 1054. We provide both inputs and targets in our dataset.

- **Are there recommended data splits (e.g., training, development/validation, testing)?** *If so, please provide a description of these splits, explaining the rationale behind them.*
  For each SatBird dataset, we provide recommended data splits for the hotspots. In order to account for spatial auto-correlation and overfitting that can arise from random splits of geospatial data, we use the `scikit-learn` DBSCAN algorithm implementation to cluster hotspots, using haversine distance. The clusters are then assigned randomly for train-validation-test, such that the proportion of hotspots in the splits corresponds to about 70%, 15% and 15% for the SatBird-USA-summer dataset. Some SatBird-USA-winter hotspots are also present in the SatBird-USA-summer ones, and we keep the assignment of hotspots from SatBird-USA-summer when splitting the SatBird-USA-winter data. More specifically, if a hotspot is in e.g. the training set of the summer dataset, it will be in the training set of the winter dataset. For the rest of the hotspots, we cluster them with the same procedure. For SatBird-USA-Kenya, the same procedure was followed. We provide more details in the "SatBird Dataset" section of our submission.

- **Are there any errors, sources of noise, or redundancies in the dataset?** *If so, please provide a description.*
  We corrected for hotspots with different hotspot ID and same latitude and longitude by merging them into one hotspot. However if multiple hotspots has the same latitude and longitude, and all checklists were reported by the same user on the same day, we filtered out these hotspots. There might be other noisy data in the eBird database that we did not correct for, which pertains to the eBird data itself. For example, there is spatial uncertainty on the locations of the reports. Also, some hotspots have very generic designation on eBird "Yellowstone park", yet are associated to a specific latitude and longitude which might not reflect the exact location at which the report was made. As for the satellite data, we selected images with less than 10% cloud coverage but there can still be presence of clouds.

## 3   Collection Process

- **What mechanisms or procedures were used to collect the data (e.g., hardware apparatus or sensor, manual human curation, software program, software API)?** *How were these mechanisms or procedures validated?* We used the Microsoft Planetary compute API to collect the Sentinel-2 data. The raw eBird checklists were downloaded directly from the eBird website. The environmental rasters were downloaded from WorldClim and SoilGrids. We also used the R package `ebirdst` [1] to extract species range maps.

- **Over what timeframe was the data collected?** *Does this timeframe match the creation timeframe of the data associated with the instances (e.g., recent crawl of old news articles)? If not, please describe the timeframe in which the data associated with the instances was created.* We extracted

complete checklists from eBird recorded from 2010 to 2023 at hotspots in the continental USA in June-July and December-January. For Kenya we extracted complete checklists from 2010 to 2023, any time in the year. For each hotspot, we associate one satellite data point (true color image + BGRNIR reflectance) in the time windows (June 1 - July 31, 2022) for USA-summer, (December 1, 2022 - January 31, 2023) for USA-winter, and (January 1, 2022 - January, 1 2023) for Kenya datasets respectively.

Bioclimatic data from WorldClim1.4 corresponds to data from 1960-1990. While this timeframe is anterior to the eBird data and satellite data used, WorldClim1.4 is a common database used by ecologists, even for modelling recent distributions.

## 4    Dataset Preprocessing, Cleaning, Labeling

- **Was any preprocessing/cleaning/labeling of the data done (e.g., discretization or bucketing, tokenization, part-of-speech tagging, SIFT feature extraction, removal of instances, processing of missing values)?** *If so, please provide a description. If not, you may skip the remainder of the questions in this section.*
  We describe the preprocessing, cleaning and labeling in the *SatBird Dataset* section of our submission in detail. Data preprocessing of the eBird data includes filtering checklists, aggregating observations of a hotspot over time, changing the format of the data, computing new statistics and correcting them using range maps from eBird among other steps. The environmental patches were extracted using the method proposed by the GeoLifeCLEF [4] challenge organizers. Our paper and the code contain description of all those procedures. We also detailed the processing of missing values in the answer to the question *What data does each instance consist of?*.
- **Was the "raw" data saved in addition to the pre-processed/cleaned/labeled data (e.g., to support unanticipated future uses)?** *If so, please provide a link or other access point to the "raw" data.* The raw eBird data and intermediate outputs of our processing pipeline were saved for internal usage. They are not made public due to storage constraints and to comply with the eBird Terms of Use. However, the raw data can be downloaded on the eBird website, and we provide all the code necessary to preprocess the data and replicate the dataset. We also only share the the environmental rasters corresponding to the hotspots of interest rather than the raw tif file from which the patches were extracted, but the raw files can be downloaded from WorldClim.

### 4.1    Dataset Use Cases

- **Has the dataset been used for any tasks already? If so, please provide a description.** The SatBird dataset has been used for the purposes of predicting bird encounter rates using machine learning in the work presented in our submission. The raw data used to obtain SatBird dataset has been used for many aplications, such as defining range maps, or predicting species abundance in the case of the eBird data, or predicting landcover for Sentinel-2 data.
- **What (other) tasks could the dataset be used for?** This task defined with this dataset could serve as an upstream task for predicting other species' distributions such as butterflies or trees.
- **Are there tasks for which the dataset should not be used?** *If so, please provide a description.* This dataset should not be used for commercial use and should respect the eBird terms of use. It should not be used for any application that could harm biodiversity.

## 5    Dataset Distribution

- **How will the dataset will be distributed (e.g., tarball on website, API, GitHub)?** *Does the dataset have a digital object identifier (DOI)?* A link to a Google Drive folder where the dataset is available on our project's website.

- **When will the dataset be distributed?** The dataset is available for download through this Google Drive link specified on the public GitHub repository, which contains the code for preparing the dataset and the benchmark, and on our project website.

- **Will the dataset be distributed under a copyright or other intellectual property (IP) license, and/or under applicable terms of use (ToU)?** *If so, please describe this license and/or ToU, and provide a link or other access point to, or otherwise reproduce, any relevant licensing terms or ToU, as well as any fees associated with these restrictions.* The SatBird Dataset is released under a Creative Commons Attribution-NonCommercial 4.0 International (CC BY-NC 4.0) License (https://creativecommons.org/licenses/by-nc/4.0/). We provide the dataset under the same terms as the eBird data which was used in this dataset. The eBird Terms of Use and the eBird Status and Trends Products Terms of Use can be found at the following links: (https://www.birds.cornell.edu/home/terms-of-use/) and Terms of Use of (https://science.ebird.org/en/status-and-trends/products-access-terms-of-use).

## 6    Dataset Maintenance

- **Who is supporting/hosting/maintaining the dataset?** We will be hosting the dataset on our institution's Google Cloud storage.

- **How can the owner/curator/manager of the dataset be contacted (e.g., email address)?** The authors can be contacted via their emails mentioned in the paper. Issues can also be opened on our public GitHub repo here.

- **Is there an erratum?** *If so, please provide a link or other access point.* Not to the best of our knowledge.

- **Will the dataset be updated (e.g., to correct labeling errors, add new instances, delete instances)?** *If so, please describe how often, by whom, and how updates will be communicated to users (e.g., mailing list, GitHub)?* We may update the dataset to have multiple satellite images corresponding a given location at several dates, instead of a single image for a given season for example. We also plan to update the environmental data as is becomes available for more recent years. In fact, we are aware that a newer version of WorldClim (version 2, which covers 1970-2000, that we used for SatBird-Kenya ) exists and we plan to update the bioclimatic data. We also plan on adding other information such as landcover or altitude. If any labeling errors our found,the dataset will be updated to correct them. The dataset updates will be reflected on the corresponding GitHub page.

- **Will older versions of the dataset continue to be supported/hosted/maintained?** *If so, please describe how. If not, please describe how its obsolescence will be communicated to users.* Should there be newer versions of the dataset, they will have the same format, and we will make sure the code associated to the project on Github can support it, and that READMEs are updated to reflect the dataset updates.

- **If others want to extend/augment/build on/contribute to the dataset, is there a mechanism for them to do so?** *If so, please provide a description. Will these contributions be validated/verified? If so, please describe how. If not, why not? Is there a process for communicating/distributing these contributions to other users? If so, please provide a description.* It will be possible to open issues and make pull requests on our dataset's companion GitHub repository.

Others are free to extend the dataset, as we provide all the necessary code for the data pipeline, as long as they share the data under the same terms as the SatBird dataset (eBird Terms of Use and the eBird Status and Trends Products Terms of Use, non commercial applications), and provided they cite all relevant sources.