# OpenReview forum: "SatBird: a Dataset for Bird Species Distribution Modeling using Remote Sensing and Citizen Science Data"
_NeurIPS.cc/2023/Track/Datasets_and_Benchmarks — NeurIPS 2023 Datasets and Benchmarks Poster_

### Official Review · Reviewer_d3tb · 2023-07-19
**Review of SatBird Dataset paper**

**Rating:** 7
**Confidence:** 3
**Clarity:** The paper is well written and easy to…

**Strengths:**

The paper has several strengths, including:

1) Well motivated and scoped. It is clear why this dataset is important (biodiversity monitoring) and how it differs from other datasets (joint modelling of bird distributions).
2) Great use of domain knowledge by including several kinds of data modalities and by utilizing range maps from eBird to apply some heuristic domain knowledge when applicable.
3) Evaluation across time of year and geographical location.

**Additional Feedback:**

In Section E of the supplementary materials a weighted CE loss is discussed be to the data imbalance. It would be fitting to reference some of the long-tailed data literature.

It is a bit of a shame that all of the results on the Winter-USA and Kenya datasets are relegated to the supplementary materials. If these could be shown in the main paper, i think the paper would have better flow and reflect the reason these splits were constructed.

**Correctness:**

In general the paper is correct. I believe the dataset is constructed in a sound way, and the benchmarks cover several types of models. I am however critical of not running any hyperparameter sweeps,

**Documentation:**

The dataset is well documented with regards to maintenance, license, availability and datasheets.

**Ethics:**

No.

**Limitations:**

The author themselves address a set of limitations such as only using a single recent image per location, and only one per season. These are the major limitations of the constructed dataset.

**Opportunities For Improvement:**

The paper could be improved in some places, including:

1) It is mentioned in the supplementary materials that no hyperparameter search was conducted (L 580-581). I'm very skeptical of this as it is unclear which hyperparameters were currently used, and the reported results may not be representative for each method. I would expect at least the learning rate to be searched.
2) It is stated that the task is treated as a "multi-output regression" task (L 82), as the label per bird species reflect the chance of encountering the species at the location. However, it is not clear from the results and discussion whether a MAE of e.g. 0.02 is large or small. It would be great if you could provide some intuition on how to interpret these values.
3) Why use the CE loss for training, when you frame the task as a regression task? While CE does support soft labels, it would be interesting to see how using losses for regression such as L1, L2, or the adaptive loss from Jon Barron (https://arxiv.org/abs/1701.03077) would affect performance.
4) There is little to no in-depth analysis of the results, on e.g. how the incorrect predictions are distributed or if there are certain patterns between incorrect predictions. It would also have been interesting to see if there was a shift in the checklists across years, i.e. did the distribution of birds at a location change over the years. If so how would you handle such a shift?

**Relation To Prior Work:**

The paper positions itself well against prior works and clear describes how it differs from such works.

**Summary And Contributions:**

The paper proposes a new dataset for bird species distribution modelling using satellite images and presence-absence observation data from a popular citizen data platform covering the last 13 years, as well as environmental data such as bioclimatic and pedologic data. The dataset is split into three scenarios: summer time in continental USA, winter time in continental USA, and Kenya. Several baselines are compared varying from Gradient Boosted Decision Trees, using pre-trained feature extractors, and training a ResNet-18 from scratch on various combinations of data.

---

> ### Author Response · Authors · 2023-08-22
> **Authors' response to reviewer d3tb**
>
> Dear reviewer,
>
> Many thanks for your time and helpful review. We address your questions in the “opportunities for improvement” below:
>
> > I would expect at least the learning rate to be searched:
>
> To address your concern regarding hyper-parameter sweeps, we have conducted a hyper-parameter search on the learning rate for our best baseline (ResNet-18 (RGBNIR+env+RM) model). We were able to find a learning rate slightly better than what we reported, which shows that we did not cherry-pick our results and we can always do better with tuning hyper-parameters.
>
> > It would be great if you could provide some intuition on how to interpret these values
>
> We have added further detailed analysis of our results and model predictions in Appendix C.2. We understand if some metrics such as MSE may not be interpretable when averaged over all data points; this is why why we also report top-10, top-30 and top-k metrics.
>
> > Why use the CE loss for training
>
> While our task is a regression task, the target is a vector of values, each with a minimum of 0 and a maximum of 1. For each species, at a given location, the target is the encounter rate of the species, which is the probability for an observer to see the species. Note that these probabilities are separate for every species: in particular, the vector of probabilities across the different birds does not sum to 1 - all the probabilities could be 0 (if there are no birds there) or could be 1 (if every observer always observed every single bird in the list). In some sense, we can consider our task as the combination of several different tasks, one for each species (where we are learning them together because the tasks share many common features). We believe that cross-entropy is suitable here because it supports soft labels as you mentioned.
>
> To verify this, we ran supplementary experiments with regression losses such as L1 Loss and L2 loss. Our results show that Cross entropy loss achieved better accuracy, and even lower MSE on our test set, when trained for the same number of epochs. We have also extended cross entropy to focal loss to help with class imbalance, but we still achieve better results with cross entropy. This matches our original hypothesis that using cross entropy is suitable for the problem definition. Results on using different loss functions are shown in Appendix B.3, Table 8.
>
> > There is little to no in-depth analysis of the results, on e.g. how the incorrect predictions are distributed or if there are certain patterns between incorrect predictions
>
> We added further analysis of the predictions in Appendix C.2. In particular, we show the distribution of the MSE by target value and the predicted values vs. the target values. We see that the model on average underestimates the encounter rates.
> We also show some example predictions in Appendix E, and see that the model can identify restricted habitat species such as the verdin.
>
> > There is little to no in-depth analysis of the results, on e.g. how the incorrect predictions are distributed or if there are certain patterns between incorrect predictions
>
> Regarding distribution shift in recent years, we did not explicitly consider it as we aggregated data over 13 years. We believe assessing the effects of rapidly changing habitats on bird distributions using our tools is extremely valuable, but that it represents a large-scale ecological study which falls outside of the scope of this project. We have noted this promising avenue for future work in the paper.
>
> > In Section E of the supplementary materials a weighted CE loss is discussed be to the data imbalance. It would be fitting to reference some of the long-tailed data literature
>
> We added further detail in Appendix D in the weighted loss paragraph to address this point.
>
> Additionally, we have moved results on SatBird-USA-Winter to the main paper as per your recommendation. We are keeping experiments on SatBird-Kenya in the appendix for the lack of space in the main text.
>
> Thank you so much for your review and please let us know if there is anything to clarify further.

---

> > ### Comment · Reviewer_d3tb · 2023-08-29
> > **Updated score**
> >
> > I would like to thank the authors for their response(s) to mine and the other reviewers comments . I find that the responses are in general satisfactory, and have increased my score to a 7. I think the paper is very interesting, focused on an important problem, and will further research within this field.

---

> > > ### Author Response · Authors · 2023-08-30
> > >
> > > Dear reviewer,
> > >
> > >  We are happy to hear that your concerns were addressed. Thank you for the time invested and your thoughtful review that helped us improve our submission.

---

### Official Review · Reviewer_ntto · 2023-07-20
**Mapping bird species to their habitats by predicting species encounter rates from satellite images.**

**Rating:** 7
**Confidence:** 4

**Strengths:**

- The presentation of the idea and how/why this dataset is useful is clear and coherent.
- Introduction and related works fairly address the main idea as well as the prior works done in the field.
- The dataset is collected from more than one region (USA, and Kenya), from different seasons (summer and winter) and for a fairly good time period (13 years).

**Additional Feedback:**

I have no additional feedback.

**Clarity:**

The paper presentation is sound and clear but the Supplementary material is not well presenting the content of the dataset.

**Correctness:**

- The Supplementary requires extensive work to describe all dataset files, as well as how they are useful in the dataset as well as the experiments conducted in the paper.

**Documentation:**

- There are links to a google drive, which contains the dataset packages as well as a link to a dropbox folder containing the codes.
- The paper does not have a landing url.

**Ethics:**

To the best of my knowledge, I don't think if there are any ethical issues with the paper.

**Limitations:**

- The article's corresponding documents available in the submission has some serious formatting issues. Submission introducing new datasets must include the supplementary materials (as one separate PDF); however, this article contains 3 different files as the supplementary material.
- The submission should not include Appendix.
- The supplementary file must include information about accessing the dataset so the authors should have all supplementary material in just one PDF file.
- I consider the file Datasheet_SatBird.pdf as the supplementary material of the paper and review based on this file. The presentation of the dataset packages lacks clarity and completion. According to the package of the Kenya,  there are 8 dataframe (.csv), 1 text (txt), as well as 4 folders. These files and folders are not introduced individually with their names and details of their contents as well as some illustrations for example for the images. Therefore it is confusing when one extracts the compressed file.
- Moreover, it is not clear what we should expect from each file or folder, for example, how the images look like as there are no samples of images and detailed descriptions of other files in the supplementary. When I opened images available in the folder "images" of the Kenya packages, I just found black images. Is that correct and should I expect this? Nothing is visible in these images!
- These issues also makes the reproduciblity challenging and difficult.

**Opportunities For Improvement:**

- The dataset could increase in size and also the number of species.
- The number of regions where the data is collected could increase.

**Relation To Prior Work:**

The paper presents relation to the prior works in introduction and related works, which is consistent with the main idea of the paper and it is within the scopes of the conference.

**Summary And Contributions:**

- The article presents SatBird, a dataset used to predict encounter rates of birds from
remote sensing images and environmental data with labels derived from eBird observation reports,
with the continental USA and Kenya as regions of interest.

- The main idea of the paper is useful for understanding the distribution of birds and their habitats by applying modern technologies rather than traditional methods used in ecology for species distribution models (SDMs).

- It also helps detecting the changes in birds distributions caused by climate changes, which is a common cause for biodiversity loss.

---

> ### Author Response · Authors · 2023-08-22
> **Authors' response to reviewer ntto**
>
> Dear reviewer,
>
> Thank you very much for your review. We really appreciate that you took the time to look at our dataset package and help us make it easier to use.
>
> * We would like to reiterate that our code and data pipeline will be available to anyone to expand the dataset to other regions or seasons.
> Thank you for highlighting areas of improvement for our submission, and we re-organized the supplementary material for more clarity accordingly, merging everything (appendix + datasheet) into one pdf file instead of a zipped file.
> * We also added more detail on how to read the data and the different files format, and improved the presentation of the data packages in the Datasheet.
> * We plan on adding a landing url for our paper (to a github repository) when this project is made public.
>
> To address more specifically some of your questions:
> > What to expect inside the compressed file:
>
> Thank you for pointing out that introducing the files and folder individually in more detail would allow for easier reproducibility. We added this information in the Datasheet.
>
> > When I opened images available in the folder "images" of the Kenya packages, I just found black images. Is that correct and should I expect this?
>
> As specified in the dataset description the “images” are actually TIF files corresponding with the (B, G, R, NIR) bands reflectance data. (which has values 0-~15000), so you should not expect to “see” anything when opening them. The “visual” component on the other hand consists of (R,G,B) bands “visual component” data with pixel values 0-255 and you should be able to see something. We clarified this further in the Datasheet. Examples of these images can be found in Appendix E, along with their predictions.
>
> We hope that this further documentation of the dataset will make reproducibility easier.  Please let us know if anything is left unclear, and thank you again for your time.

---

> > ### Author Response · Authors · 2023-08-30
> >
> > Dear reviewer,
> >
> >  We are happy to hear that your concerns were addressed. Thank you for the time invested and your thoughtful review that helped us improve our submission.

---

### Official Review · Reviewer_rCBg · 2023-07-20
**A novel task but additional experinments and discussions**

**Rating:** 7
**Confidence:** 5
**Correctness:** It's appropriate
**Clarity:** Yes

**Strengths:**

1. The paper is well-motivated
Climate change leads to biodiversity loss, impacting ecosystem services and human well-being. Understanding global species distributions is essential for policy decisions like land use and conservation.

2. The proposed task is novel
Predicting encounter rates of bird species using remote sensing and citizen science data is new and worth exploring.

3. The evaluation metrics are well-defined

**Additional Feedback:**

I'd also suggest author to include the computation cost/time in the paper along with different methods.

**Documentation:**

Yes, it's sufficient.

**Ethics:**

No ethical concerns from me.

**Limitations:**

Yes, sufficient discussions are given.

**Opportunities For Improvement:**

W1: Sufficient discussion of related work:
While the proposed prediction tasks are based on remote sensing datasets, I'd hope to see more comparisons between the proposed data and training datasets in remote sensing like Extended Agriclture-Vision[1] or EuroSAT[2]. What makes the proposed datasets unique, and what is missing(like the temporal information)?

W2. Scale and Range of the benchmarks

While the authors propose several benchmarks based on the proposed method, the scale is relatively small. For example, the backbones are limited to ResNet18. Additionally, is it possible to take advantage of weights from other papers pretrained from Remote sensing datasets like [2,3]?

[1] Extended Agriculture-Vision: An Extension of a Large Aerial Image Dataset for Agricultural Pattern Analysis

[2] Eurosat: A novel dataset and deep learning benchmark for land use and land cover classification

[3] Seasonal contrast: Unsupervised pre-training from uncurated remote sensing data

**Relation To Prior Work:**

More discussion is expected

**Summary And Contributions:**

The authors propose a task of predicting encounter rates of bird species at specific locations using remote sensing data.
To support this task, the authors introduce SatBird, a dataset obtained from publicly available bird observation and satellite data sources.
The dataset includes sub-datasets for the USA summer (breeding season), USA winter (nonbreeding season), and Kenya (low-data regime) as an example.
The authors evaluate several popular models on SatBird and demonstrate the effectiveness of deep computer vision methods for this task.

---

> ### Author Response · Authors · 2023-08-22
> **Authors' response to reviewer rCBg**
>
> Dear reviewer,
>
> Many thanks for your insightful review. We thank you for recognising the novelty and ecological importance of this machine learning task. We hope that the revisions made to our submission address the points you raised.
>
> > W1: Sufficient discussion of related work:
>
> Thank you for pointing out relevant literature and we have updated our Related Work (Section 3) of the main paper to include further details and references to remote sensing datasets as suggested.
>
> > W2. Scale and Range of the benchmarks
>
> Upon your suggestion, we utilized Seasonal Contrast (SeCo) pretrained model weights, and performed an additional experiment with the benchmark model that performed best (ResNet-18 with RGBNIR+env+RM) with weighted transferred from SeCo. We report the results in Appendix B.2. We found that initializing the model with SeCo gave close results to initializing the model with ImageNet weights.
>
> The computation time for our baseline ResNet models was reported in the Experiments section, Appendix B, and we added the information for the Satlas and SatMAE model as well.
>
> Please let us know if you would like any additional information and thank you again for your time.

---

> > ### Author Response · Authors · 2023-08-30
> > **Following up on our rebuttal**
> >
> > Dear reviewer,
> >
> > We understand the timeline for reviews is tight, but we would love to discuss with you any concerns that may remain open after our response. We would be grateful if you could confirm that you have read our responses and let us know whether we have successfully addressed the concerns or if there are further points you would like to discuss.
> >
> > Thank you again for taking the time for this review.

---

> > > ### Comment · Reviewer_rCBg · 2023-08-30
> > >
> > > I appreciate the author's efforts in their rebuttal. The additional experiments/discussions have addressed most of my concerns. So I'll raise my rate.

---

### Official Review · Reviewer_Sbqv · 2023-07-20
**Interesting Dataset but need more information on the dataset and benchmarks results**

**Rating:** 6
**Confidence:** 3

**Strengths:**

I like the motivation of the dataset and the challenge of the task (modelling the species distribution of birds) the dataset is attempting to answer. It is clear that the authors have put in substantive amount of work in creating the dataset, and the approach developed by the authors will be helpful for inventorying  bio-diversity especially of species that are threatened. I also like that the dataset covers multiple geographic and seasons, and while it would be good to see a longitudinal study over several years to study the health of the conservation of the species. this dataset is a good start to build future datasets that can address that problem.

**Additional Feedback:**

Please look at the Opportunities for Improvements and Correctness for suggestions to improve the paper and improve confidence in the benchmarks.

**Clarity:**

The paper is well written and somewhat easy to follow, with no obvious grammatical mistakes. However do note that English is not my first language

**Correctness:**

The benchmark could do with a lot more detail. In particular i was hoping that the authors provided some break down in the distribution of the regression error by predicted/target value, species, and so forth. It would be good to
While the top-10/30/k classification accuracies are helpful, the single MAE+MSE value provided in Tables 2, 5-9 to me is not a sufficient measure to benchmark what is inherently a regression task.

**Documentation:**

The documentation seems sufficient to me

**Ethics:**

No, i do not suspect that there are ethical concerns with this submission

**Limitations:**

I am unclear how reliable the ground truths estimates of the base encounter rates are. I have a passing knowledge of this area however am not familiar with the reliability of the estimates provided by eBirds or if there have been studies regarding that. It would be good if the authors could provide some resources regarding that, or point to methods the estimates account for observational biases.

**Opportunities For Improvement:**

The paper could do with quite a bit more detail beyond what was provided in the main paper and the It would be helpful if the authors provide some descriptive statistics of the dataset esp of the target variable (namely the encounter rate), i/e the histogram of the encounter rate distribution of some of the species/ or histogram of the species by encounter rate.

**Relation To Prior Work:**

While the paper did compare their work to other similar datasets of different species of animals ie sharks, butterflies, and plants. I would have preferred if the authors also mention the other approach to estimating bird density using bird acoustics and the limitations of those approaches vs satellite imagery.

**Summary And Contributions:**

This paper describes SatBird, a satellite based hyperspectral image dataset (provided by Sentinel-2) that combines observational data collected from citizen science (eBird), environmental bioclimatic features extracted from GeoLifeCLEF and SoilGrids in order to estimate the encounter rates. The dataset covers 2 seasons namely summer and winter in the US and one season in Kenya. While the authors provided some benchmarks of some SOTA methods on the dataset, I was expecting more details and breakdown especially on the main task of encounter rate estimations.

---

> ### Author Response · Authors · 2023-08-22
> **Authors' response to reviewer Sbqv**
>
> Dear reviewer,
> Thank you for your helpful comments. Upon your suggestion, we added descriptive statistics of the dataset in the Appendix A, providing a histogram of the encounter rate distribution of some species across hotspots, as well as a histogram of the species by mean encounter rate across training hotspots. We also added a more detailed analysis of the results in Appendix C.2, in order to make our benchmark stronger. Below, we address the comments for each topic section of the review. Please let us know if there is anything you would like more clarification on.
>
> > Limitations:
>
> While the reliability of the ground truth estimates of the encounter rates from eBird data has, to our knowledge, not been systematically assessed, previous work has shown that eBird data is reliable and accurate. In the case of estimating population trends, it has been shown to give estimates that only differ marginally from estimates using formal surveys [1]. eBird data has also been used to inform policy directly in a study that  compared using eBird data with 4 other bird survey data sources and found that eBird data gave the best estimate of avian abundance and distribution [2] . While we did not take into account the reporting patterns of birders (we aggregated data per hotspot), [3] is an example of previous work that demonstrates how estimating the observer expertise can improve species distributions from eBird data. We added more information about this in the paper based on your feedback.
>
> [1] Horns, Joshua J., Frederick R. Adler, and Çağan H. Şekercioğlu. "Using opportunistic citizen science data to estimate avian population trends." Biological conservation 221 (2018): 151-159.
>
> [2] Ruiz-Gutierrez, V., E. Bjerre, M. Otto, G. Zimmerman, B. Millsap, D. Fink, E. F. Stuber, M. Strimas-Mackey, and O. J. Robinson. (2021). A pathway for citizen-science data to inform policy: a case study using eBird data for defining low-risk collision areas for wind energy development. Journal of Applied Ecology.
>
> [3] Johnston, A., Fink, D., Hochachka, W. M., & Kelling, S. (2018). Estimates of observer expertise improve species distributions from citizen science data. Methods in Ecology and Evolution, 9(1), 88-97.
>
> > Opportunities for improvement
>
> We thank you for suggesting to further describe the dataset. We added descriptive statistics of the target values (the encounter rates) in Appendix A.
>
> > Correctness
>
> We thank you for highlighting ways to improve our benchmark. We are adding further analysis of the results and provided the following in Appendix C.2:
> - Distribution of the regression error by predicted/target values (Fig. 10)
> - Visualization of the predicted vs target values
> - Analysis of the performance on a species level in Tables 11-12.
>
> For the best model of our benchmark on the SatBird-USA-summer dataset, we find that the best predicted species according to MSE are species which are reported in few hotspots (< 10). This is not surprising as for those species, the model could always predict 0 and have low MSE for these species. This reflects the zero-inflated nature of our targets and supports the need for other metrics than MSE and MAE that we highlight in Section 5.3.  We also find that the model on average slightly underestimates the encounter rates (Appendix C.2).
>
> > It would be good to While the top-10/30/k classification accuracies are helpful, the single MAE+MSE value provided in Tables 2, 5-9 to me is not a sufficient measure.
>
> Regarding the evaluation metrics, MSE and MAE are usually reported for regression tasks but indeed do not fully reflect performance in our case which is why we also considered top-10/30/k accuracies. Please let us know if the addition of further analysis addresses your point, or if there is something more in particular that would make our benchmark stronger. It seems one of the sentences in the review was cut off and we would be happy to know what your suggestion was to further improve our submission.
> Additionally, we also show examples of predictions on sample locations in Appendix E. We can see that the model is able to identify the presence of rare birds such as the verdin, which is only found in specific habitats (mainly desert shrublands) .
>
> > Relation to prior work
>
> We added a mention of work using bioacoustics for bird distribution modeling in the Related Work section.
>
> Thanks again for your time and feedback.

---

> > ### Author Response · Authors · 2023-08-30
> > **Following up on our rebuttal**
> >
> > Dear reviewer,
> >
> > We understand the timeline for reviews is tight, but we would love to discuss with you any concerns that may remain open after our response. We would be grateful if you could confirm that you have read our responses and let us know whether we have successfully addressed the concerns or if there are further points you would like to discuss.
> >
> > Thank you again for taking the time for this review.

---

> > > ### Comment · Reviewer_Sbqv · 2023-08-30
> > > **Response to rebuttal**
> > >
> > > Thanks for the response to my review, it does address some of my concerns regarding the completeness of the evaluation. I am amenable to revise my ranking upwards.

---

### Official Review · Reviewer_8dHr · 2023-07-21
**Some good points, but missing biological aspects**

**Rating:** 5
**Confidence:** 3
**Clarity:** Yes, clear story,

**Strengths:**

- Good agglomeration of different dataset
- Good variety of tested machine learning models for the specific task.
- Well written article, clear story, and good definition of the dataset task (in the main article and the appendix).
- Code is well written with comments.



**Additional Feedback:**

This work is interesting and impressive. However, there are some corrections needed in the manuscript and maybe the dataset to make it relevant to the biologists and decision makers.

**Correctness:**

Although I haven't replicated the authors results, however the results seems plausible.
- Lots of zero (0.0) in Table 2.
- The provided source code is mostly for a ViT transformer, not reported in the paper (?).

**Documentation:**

Yes. However, could add some information about the satellite imagery and Worldclim licenses.

**Ethics:**

No problem

**Limitations:**

- Need a better biological review and some validation of some data about bird habitat from the literature (not just citizen science). Also, some of the citations (e.g. 26, 28) seems out of scope with the main topic.
- No information for specific bird predictions, limiting the interpretation of the results (e.g. is 46.2% accuracy good? What are we missing in the prediction?)
- Miss some meta-data for the land use, land cover associated with the satellite imagery.
- The included satellite imagery (Sentinel-2) is limited to 4 channels.
- No clear explanation why Kenya was chosen in Africa, over other countries.



**Opportunities For Improvement:**

- Add a research hypothesis
- Add some citations about the use of satellite imagery for avian habitat relationship (e.g. https://doi.org/10.1080/01431160512331338041, https://doi.org/10.1002/eap.2157)
- Expend the dataset to bird of North America (not just USA) by focusing on indicative species.
- Could the authors elaborate on the avian species presence-absence instead of relative abundance?
- One of the interesting aspect of the datasets is the possibility of predicting avian species habitat for conservation priorities (conclusion section). However, this aspect is not demonstrated. Since we have past satellite imagery and temperature, could the authors show some predictions from rapidly changing habitats?
- Could the authors present some evidence about the usefulness of the pedology information in predicting bird habitats? It seems that this information is mostly important for the distribution of tree/plant species. What is the gain of adding this layer instead of data from tree/plant species (e.g. from the spatial database of planted trees?).
-Would the model be better at predicting only one bird instead of the whole distribution?
- Figure 2 legend is very small and the selected color not helpful to color blind scientists.
- Add information about the deep learning library used for the ResNet-18 (pytorch? keras? tf?)


**Relation To Prior Work:**

This section might be reframed since there are literature on the use of deep learning for bird's habitat prediction, however, most of this section focusses on citizen science and deep learning competition. Better explication of some model for predicting presence-absence, bird habitats, etc. might be useful.

**Summary And Contributions:**

In this manuscript, the authors combined Sentinel-2 data, soil information (pedology, soilgrids) and bioclimatic information to the ebird dataset. The goal of the authors is to predict the presence or absence of bird species in the USA and Kenya. They showed the performance of different classifiers (SATLAS, SatMAE, MOSAIKS, ResNet-18) at predicting the species observation likeliness (based on the ebird complete checklist data). They clearly mention in the conclusion some drawback of their dataset: 1) single satellite image for each hotspot, and 2) the aggregation of the ebird data for multiple years. One of the first questions arising from this dataset is the intended use: the mapping of birds to their habitat. Unfortunately, the dataset can only do correlation about the geographic location and possibly weather information although with some biases, the limitation being that no information about the composition of the habitat is included in the dataset (e.g. % water, % crop, % forest, forest type). Furthermore, being limited to the USA and Kenya, the usefulness of the dataset seems limited for conservation studies.

---

> ### Author Response · Authors · 2023-08-22
> **Authors' response to reviewer 8dHr**
>
> Dear reviewer,
> We thank you for your thoughtful review and insights; we believe the revisions made upon your feedback makes it stronger. In particular, we add references about the use of satellite imagery for avian habitat modeling in the main paper, as well as further analysis and interpretation of the results obtained with our models (including details on a species-level), and expand on choices made in the construction of our dataset in Appendix A. We also changed the colors of Fig. 2. Below, we reply to your questions:
>
> > Research hypothesis
>
> Our hypothesis is that we can approximate habitat information from globally available sources of satellite imagery. Complementing work such as Cole, et al (ICML 2023), which is focused on inferring large-scale maps of species distribution from pointwise observations, our goal is to better predict fine-scale species occurrence patterns (e.g., when a certain species is found only in forests of a certain age and species composition), beyond what can be obtained with environmental data which is traditionally used in SDM models. We would also like to emphasize that while we are here releasing Satbird-USA and Satbird-Kenya, our code and data pipeline will be available for anyone to expand Satbird beyond USA and Kenya.
>
> > Expanding the dataset to birds of North America
>
> While our geographical area of focus for the Satbird-USA datasets is the USA, the 684 considered species represent the regularly occurring (ABA Codes 1 and 2) avifauna of the entire “ABA Area” (North America north of Mexico). As noted above, we encourage the use of our openly available code and data pipeline to expand Satbird to other regions.
> > avian species presence-absence vs. relative abundance
>
> We chose to predict encounter rates derived from presence-absence data both for the ecological value of this measure and because of the reliability of presence-absence checklists for validating our approach. Data for assessing relative abundance is less widely available in the eBird database, since some complete checklists do not include the count of individuals of a given species.
> > could the authors show some predictions from rapidly changing habitats
>
> By predicting which species can be found currently across large geographical areas, SatBird can already allow policymakers to prioritize areas of land for present habitat conservation initiatives. We believe assessing the effects of rapidly changing habitats on bird distributions using our tools is extremely valuable but that it represents a large-scale ecological study which falls outside of the scope of this project. We have noted this promising avenue for future work in the paper.
> >  usefulness of the pedology info. in predicting bird habitats
>
> We have not found specific evidence about the usefulness of pedology data for bird habitats. However, it has been shown that adding soil data can improve the prediction of butterfly distributions compared to just using climate information (Titeux, et al. J. Biogeography 2009).
> > On metadata for the land use, land cover associated with the satellite imagery
>
> We previously experimented with land use on a subset of Satbird-USA-summer using ResNet-18 model with environmental, RGBNIR, and land use as input but did not observe much effect from adding the land use information and therefore did not include it in our dataset. We strived to use only publicly and globally available data for this project; therefore, we used the 10m resolution Esri 2020 Land Cover collection which only has 10 classes. We believe that coarse classification likely limited the added value of the land use data.
> > Could the model be better at predicting only one bird instead of the whole distribution?
>
> We have explicitly replied to this question, with empirical work, in Appendix B.4
>
> > Why Kenya?
>
> We included the US since it has the highest number of eBird submissions and therefore the data is extremely broad and reliable. With Kenya, we intended to complement our choice of the US, in particular motivated by the following:
> - Much lower but still significant amounts of bird observation data
> - Significantly different biomes from the US
> - Extremely high bird diversity in a concentrated area
> - A location with large numbers of nonmigratory species, unlike the US (allows the use of a single dataset without splitting into breeding / non breeding)
> - National priority on conservation, making additional data on avifauna especially valuable (GoK. Kenya Vision. 2030)
>
> Additionally, we confirm that the shared code includes implementations to all the models reported, not only the ViT. We updated the README of our code to indicate where to find the different models. For the 0.0 std values in Table 2, results (MSE/MAE) obtained with some of the baselines were indeed numerically very close across different seeds, but we fixed this by increasing the precision.
>
> Please let us know if you would like further clarification. Thank you for helping us improve our submission.

---

> > ### Author Response · Authors · 2023-08-30
> > **Following up on our rebuttal**
> >
> > Dear reviewer,
> >
> > We understand the timeline for reviews is tight, but we would love to discuss with you any concerns that may remain open after our response. We would be grateful if you could confirm that you have read our responses and let us know whether we have successfully addressed the concerns or if there are further points you would like to discuss.
> >
> > Thank you again for taking the time for this review.

---

### Author Response · Authors · 2023-08-22
**Summary response to all reviewers with outlined changes**

We thank the reviewers for their time and detailed feedback.
We really appreciate that all reviewers value the contribution of the SatBird dataset to the biodiversity community and highlight the good definition and motivation of the dataset task.  We thank the reviewers for putting forward that the SatBird task is “new and worth exploring” [rCBg19], for praising our “good variety of tested machine learning models for the specific task” as well as the “well written article and code” [8dHr21], noting the “great use of domain knowledge with range maps'' [d3tb19], and for recognizing the effort put into building a dataset covering multiple geographic areas and seasons [ntto19, Sbqv20].

We reply to each reviewer’s specific questions in the individual review threads.
In this overall response, we explain the revisions to the paper, taking into account the reviewers’ thoughtful feedback.

We make the following changes to our submission:
- Expand relation to prior work by adding more references to work on bird habitat prediction using deep learning, and datasets in remote sensing.
- Move results on the SatBird-USA-winter from the Appendix to the main paper
- Add descriptive statistics of the targets (encounter rates) in Appendix A.
- Add further details on the deep learning library used in our implementation, as well as, computation time and cost of our models, in Appendix B.
- Expand the analysis and interpretation of the results obtained with our benchmark model in Appendix C.2, and show sample predictions in Appendix E.
- Re-organize supplementary material in a single file, and clarify how to use the dataset

We also conducted the following experiments in response to reviewer suggestions:
- Hyper-parameter search on the learning rate for the ResNet-18 (RGBNIR + env + RM) (Appendix B.1)
- Fine-tuning a model initialized with pretrained weights from Seasonal Costrast [1]  (Appendix B.2)
- Trying different loss functions such as L1 loss, L2 loss and focal loss. (Appendix B.3)
- Comparing the performance of 20 single species models vs. a model trained on all 20 species simultaneously vs. the performance of the main model, which is trained to predict 684 species, evaluated on the same subset of 20 species  (Appendix B.4)

We also did a few minor changes:
- Integrate errata in Appendices A.1 in B.1 into the paper itself (SatBird-USA-winter dataset train/val/test repartition and results for the environmental baseline on the test set of the SatBird USA-summer dataset).
- Change colors of Fig. 2 to make it colorblind-friendly.


[1] Seasonal contrast: Unsupervised pre-training from uncurated remote sensing data, Oscar Manas, Alexandre Lacoste, Xavier Giro-i-Nieto, David Vazquez, Pau Rodrigue, 2021 IEEE/CVF (ICCV)

---

### Decision · Program_Chairs · 2023-09-22

**Decision:**

Accept (Poster)

**Comment:**

The reviewers generally agree on the following strengths:

- this new dataset is well-motivated and has potential to have impact for biodiversity monitoring.
- the dataset differs from existing datasets, e.g., joint modeling of bird distributions.
- it combines a wide range of times and geographical location, makes great use of domain knowledge, and has a good variety of tested machine learning models.
- the paper is well written, especially after more details were added during the rebuttal.

The reviewers had weaknesses that were generally addressed adequately during the rebuttal. For example,

- the authors added significantly more details and descriptions to various parts of their submission.
- added new experiments during the rebuttal: HPO for baselines, fine-tuning pretrained models, other loss functions, and have compared a baseline of using 20 single species models vs. one 20 species model.
- The only reviewer who gave a 5, was also the only one who did not reply to the rebuttal, having this review being is weighted less. I believe the authors’ response was adequate.